# AMPK-dependent activation of the Cyclin Y/CDK16 complex controls autophagy

Marc Dohmen[1,8,11], Sarah Krieg [1,11], Georgios Agalaridis[1,9], Xiaoqing Zhu[2], Saifeldin N. Shehata[3], Elisabeth Pfeiffenberger[4], Jan Amelang [1], Mareike Bütepage[1], Elena Buerova[1], Carolina M. Pfaff[1,10], Dipanjan Chanda[2], Stephan Geley[4], Christian Preisinger[5], Kei Sakamoto[3,6], Bernhard Lüscher [1,12✉], Dietbert Neumann [2,7,12✉] & Jörg Vervoorts [1,12✉]

The AMP-activated protein kinase (AMPK) is a master sensor of the cellular energy status that is crucial for the adaptive response to limited energy availability. AMPK is implicated in the regulation of many cellular processes, including autophagy. However, the precise mechanisms by which AMPK controls these processes and the identities of relevant substrates are not fully understood. Using protein microarrays, we identify Cyclin Y as an AMPK substrate that is phosphorylated at Serine 326 (S326) both in vitro and in cells. Phosphorylation of Cyclin Y at S326 promotes its interaction with the Cyclin-dependent kinase 16 (CDK16), thereby stimulating its catalytic activity. When expressed in cells, Cyclin Y/CDK16 is sufficient to promote autophagy. Moreover, Cyclin Y/CDK16 is necessary for efficient AMPK-dependent activation of autophagy. This functional interaction is mediated by AMPK phosphorylating S326 of Cyclin Y. Collectively, we define Cyclin Y/CDK16 as downstream effector of AMPK for inducing autophagy.

[1] Institute of Biochemistry and Molecular Biology, Medical School, RWTH Aachen University, 52074 Aachen, Germany. [2] CARIM School for Cardiovascular Diseases, Maastricht University, P.O. box 616, 6200 MD Maastricht, The Netherlands. [3] Nestlé Research, EPFL Innovation Park, 1015 Lausanne, Switzerland. [4] Division of Molecular Pathophysiology, Biocenter, Innsbruck Medical University, Innrain 80/82, 6020 Innsbruck, Austria. [5] Proteomics Facility, Interdisciplinary Center for Clinical Research (IZKF) Aachen, Medical School, RWTH Aachen University, 52074 Aachen, Germany. [6] Novo Nordisk Foundation Center for Basic Metabolic Research, University of Copenhagen, Copenhagen, Denmark. [7] Department of Pathology, University Medical Center Maastricht, 6229 HX Maastricht, The Netherlands. [8] Present address: Center for Translational & Clinical Research Aachen (CTC-A), Medical School, RWTH Aachen University, 52074 Aachen, Germany. [9] Present address: Miltenyi Biotec GmbH, Friedrich-Ebert-Straße 68, 51429 Bergisch Gladbach, Germany. [10] Present address: AstraZeneca GmbH, Tinsdaler Weg 183, 22880 Wedel, Germany. [11] These authors contributed equally: Marc Dohmen, Sarah Krieg. [12] These authors jointly supervised this work: Bernhard Lüscher, Dietbert Neumann, Jörg Vervoorts. ✉email: luescher@rwth-aachen.de; d.neumann@maastrichtuniversity.nl; jvervoorts-weber@ukaachen.de

The AMP-activated protein kinase (AMPK) plays an important role in balancing energy homeostasis through phosphorylating downstream targets. AMPK promotes ATP production by activating catabolic processes and limits ATP consumption by inhibiting anabolic processes under energy stress[1]. AMPK is a heterotrimeric complex with a catalytic α subunit and regulatory β and γ subunits[2]. Increased cellular AMP concentration, e.g., in response to nutrient starvation, is sensed by the γ subunit, promoting a conformational change that stimulates AMPK activity. AMP or ADP binding facilitates phosphorylation of the α subunit at threonine 172 (T172). Liver kinase B1 (LKB1) is the major T172 kinase in response to energy stress and T172 phosphorylation activates AMPK[3,4].

AMPK has been linked to the regulation of macroautophagy (hereafter autophagy), an intracellular degradation and recycling mechanism activated in response to nutrient deprivation, viral infection, organelle damage or formation of protein aggregates. During autophagy a double membrane is formed to surround and isolate cytoplasmic content, which is then delivered to lysosomes for degradation and recycling. A cascade of conserved protein complexes, including autophagy-related proteins (ATGs), regulates the different steps of autophagy[5]. The first two complexes involved in integrating signals for the induction of autophagy are the ATG1/ULK1 complex and the class III phosphatidylinositol-3-kinase (PI3K) complex. The latter consists of the catalytic subunit vacuolar protein sorting mutant 34 (VPS34), the regulatory subunits Beclin1, p150, and ATG14. These complexes are connected to the upstream regulators mammalian target of rapamycin complex 1 (mTORC1) and AMPK[6]. Under nutrient-rich conditions AMPK is suppressed, while mTORC1 is active and inhibits the ATG1/ULK1 complex and thus autophagy. In turn, AMPK is activated in response to nutrient deprivation or other autophagy-promoting cues. Subsequently, it phosphorylates and activates the mTORC1 inhibitor tuberous sclerosis 2 (TSC2) and phosphorylates the mTORC1 subunit Raptor, which together inhibits mTORC1[7,8]. Moreover, AMPK activates the ATG1/ULK1 complex by phosphorylating ULK1[9,10]. AMPK also stimulates the autophagy-specific functions of the class III PI3K complex by phosphorylating Beclin1 and VPS34[11]. Of note is that the mechanisms of these activations by AMPK as well as the full spectrum of AMPK substrates involved in autophagy are only partially known. The cyclin-dependent kinase (CDK) 16 belongs to the PCTAIRE family consisting of CDK16, 17 and 18, which are characterized by a highly conserved kinase domain closely related to those of cell cycle regulating CDKs[12]. In addition to CDK16 (PCTAIRE1), Cyclin Y also interacts with and activates CDK14 (PFTAIRE1)[13]. The CDK16-18 kinase domains are flanked by unique N- and C-terminal extensions[13]. Binding of Cyclin Y to CDK16 requires both the N- and C-terminal regions of CDK16[14] and, in addition, the phosphorylation of S100 and S326 of Cyclin Y[15].

To understand the molecular functions of AMPK in more detail, we perform a kinase substrate screen on protein microarrays and identify the Cyclin Y/CDK 16 complex as AMPK substrate. We identify Cyclin Y S326 as the key target of AMPK, thereby promoting the interaction with CDK16 and stimulating the kinase activity of CDK16. Our findings further define the active Cyclin Y/CDK16 complex as crucial promoter of autophagy and cellular effector of AMPK.

## Results

**The Cyclin Y/CDK16 complex is an AMPK substrate**. AMPK is an energy sensor and implicated in processes that include autophagy, control of cell polarity and cell growth. To understand the AMPK-dependent regulation of these processes in more detail, we searched for AMPK substrates by performing in vitro kinase assays on a protein microarray (ProtoArray) with over 9000 different human proteins (Fig. 1a). Active trimeric AMPK-α1β1γ1 was obtained by bacterial co-expression with the activating LKB1 from a single polycistronic vector[16]. The arrays were incubated with radiolabeled ATP in the presence or absence of AMPK. The signals on the array were evaluated according to the manufacturer's protocol to define putative targets (Supplementary Data 1). In total 63 proteins fulfilled the set criteria, of which four proteins are known AMPK targets[17] (Supplementary Table 1). We identified both subunits of the Cyclin Y/CDK16 kinase complex as AMPK substrates (Fig. 1b). The phosphorylation of Cyclin Y and CDK16 was confirmed in in vitro kinase assays with recombinant GST-CDK16 and Cyclin Y-His$_6$ and active AMPK-α1β1γ1 (Fig. 1c). Hence we tested whether AMPK affected the interaction of Cyclin Y with CDK16 and the catalytic activity. Cyclin Y and CDK16 were co-expressed in HeLa cells. These cells are deficient for LKB1 but respond to treatment with a combination of AICAR and A769662, which activates AMPK[18,19]. Notably, the combined AICAR/A769662 treatment activates AMPK independently of T172 phosphorylation of the α-subunit[20]. Indeed, AICAR/A769662 stimulated the phosphorylation of acetyl-CoA carboxylase (ACC) (Fig. 1d), a known AMPK substrate[21]. AMPK stimulation enhanced the interaction between Cyclin Y and CDK16, as determined by co-immunoprecipitation, by 2-fold (Fig. 1d, e) and increased kinase activity of the Cyclin Y/CDK16 complex towards myelin basic protein (MBP) (Fig. 1d). Therefore, the data suggest that the in vitro identified AMPK substrates Cyclin Y and CDK16 are also regulated by AMPK in cells.

**AMPK phosphorylation of Cyclin Y controls CDK16 activity**. To determine whether the cellular regulation of Cyclin Y/CDK16 depends on direct phosphorylation by AMPK, we mapped the AMPK phosphorylation sites. For CDK16 we identified eight potential in vitro AMPK phosphorylation sites using mass spectrometry (Supplementary Table 2). These sites could not be verified in cells after immunoprecipitation of GFP-tagged CDK16 by mass spectrometry because the sites either did not respond to AMPK activation by a combined AICAR and A769662 treatment or were not phosphorylated (Supplementary Table 2 and Supplementary Data 2 with the MaxQuant output of the phosphosite analysis). In parallel, we generated and tested phosphorylation site-specific antibodies for S12, S65, S119, S153, S155, and S461 of CDK16. While phosphorylation of CDK16 by AMPK in vitro was detectable for S12, S153, S155, and S461 (Supplementary Fig. 1a), phosphorylation was unaffected in cells upon AMPK activation. The phosphorylation of S155 was not detectable in cells with a phospho-specific antibody and the phospho-S461 antibody did not recognize CDK16 in a phospho-specific manner (Supplementary Fig. 1b). Also the phosphorylation of S65 and S119 did not change after AMPK activation in cells (Supplementary Fig. 1c). In addition, an siRNA-mediated knockdown of AMPK had no effect on S65 and S119 phosphorylation (Supplementary Fig. 1d). Thus, none of the in vitro identified AMPK phosphorylation sites of CDK16 appear to be prominent AMPK substrates in cells.

Therefore, we focused on Cyclin Y. We identified S326 as in vitro AMPK phosphorylation site by mass spectrometry (Supplementary Fig. 2a and Supplementary Table 2). This site matches the AMPK consensus motif generated from 64 validated AMPK targets[17]. Furthermore, the site is evolutionarily conserved between mammals, lower vertebrates and arthropods (Supplementary Fig. 2b). We confirmed the AMPK-dependent phosphorylation of Cyclin Y in vitro using a phospho-S326 specific

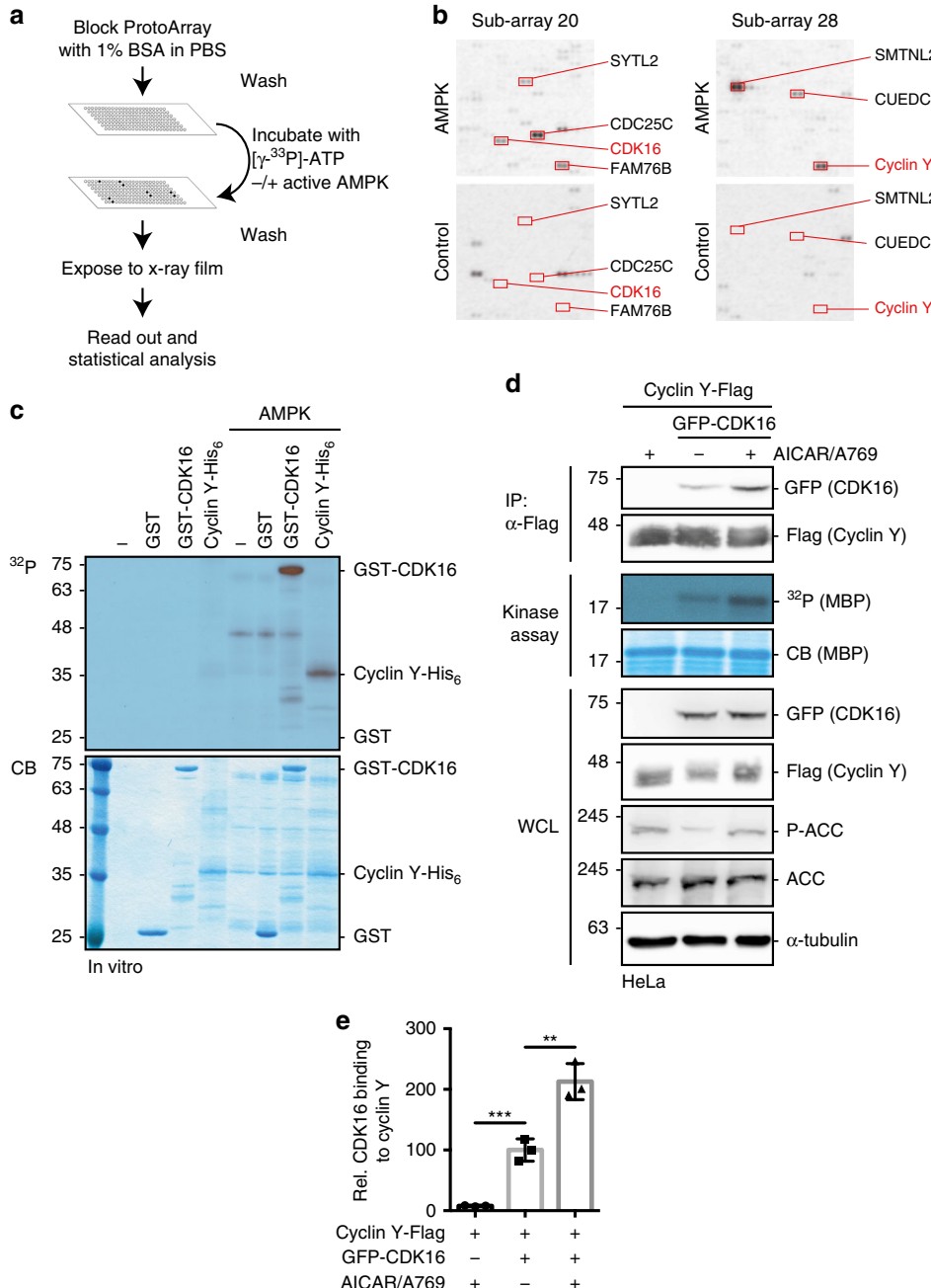

**Fig. 1 Protein microarray screen for the identification of AMPK substrates. a** Schematic representation of the ProtoArray based screen with approximately 9000 human proteins using AMPK (see also Supplementary Data 1). **b** Details of two sub-arrays incubated with or without AMPK with marked substrates are shown. **c** GST-CDK16, Cyclin Y-His$_6$ and GST were incubated in the presence of [γ-$^{32}$P]-ATP with AMPK. Phosphorylation was determined by autoradiography ($^{32}$P, top). Proteins were visualized by Coomassie blue staining (CB, bottom; $n = 2$). **d** HeLa cells were transfected with vectors expressing GFP-CDK16 and Cyclin Y-Flag and treated for 1 h with 0.5 mM AICAR/50 μM A769662 (A769) as indicated. Cyclin Y-Flag was immunoprecipitated with Flag antibodies (IP) and immunoblotted against CDK16 and Cyclin Y or used for in vitro kinase assays with myeloid basic protein (MBP) as substrate. Autoradiographs ($^{32}$P) and Coomassie blue staining (CB) of MBP are displayed. Whole cell lysates (WCL) were immunoblotted with the indicated antibodies ($n = 3$). **e** Quantification of CDK16 co-immunoprecipitated with Cyclin Y. Statistical significance was measured via unpaired and two-tailed Student's $t$-tests and is presented as follows: $**p < 0.01$, and $***p < 0.001$. All error bars indicate SD ($n = 3$; Cyclin Y + AICAR/A769 vs. Cyclin Y/CDK16: t = 8.719, df = 4; Cyclin Y/CDK16 vs. Cyclin Y/CDK16 + AICAR/A769: t = 5.595, df = 4). n biological independent replicate. SD standard deviation. Source data are provided as a Source Data file.

antibody (Fig. 2a)[15]. In this assay the phospho-S326 antibody showed a more then 40-fold increased specificity for the S326 phosphorylated Cyclin Y compared to the unphosphorylated or the S326A mutant Cyclin Y. To determine the specificity, we used a longer exposure of the phospho-S326 blot where binding to the

unphosphorylated and phospho-mutant Cyclin Y was visible (Fig. 2a). In a next step the S326 phosphorylation was validated in HeLa cells. S326 phosphorylation increased in response to AMPK activation with the AMPK substrates ACC and Raptor for control (Fig. 2b). This was accompanied by Cyclin Y phosphorylation at

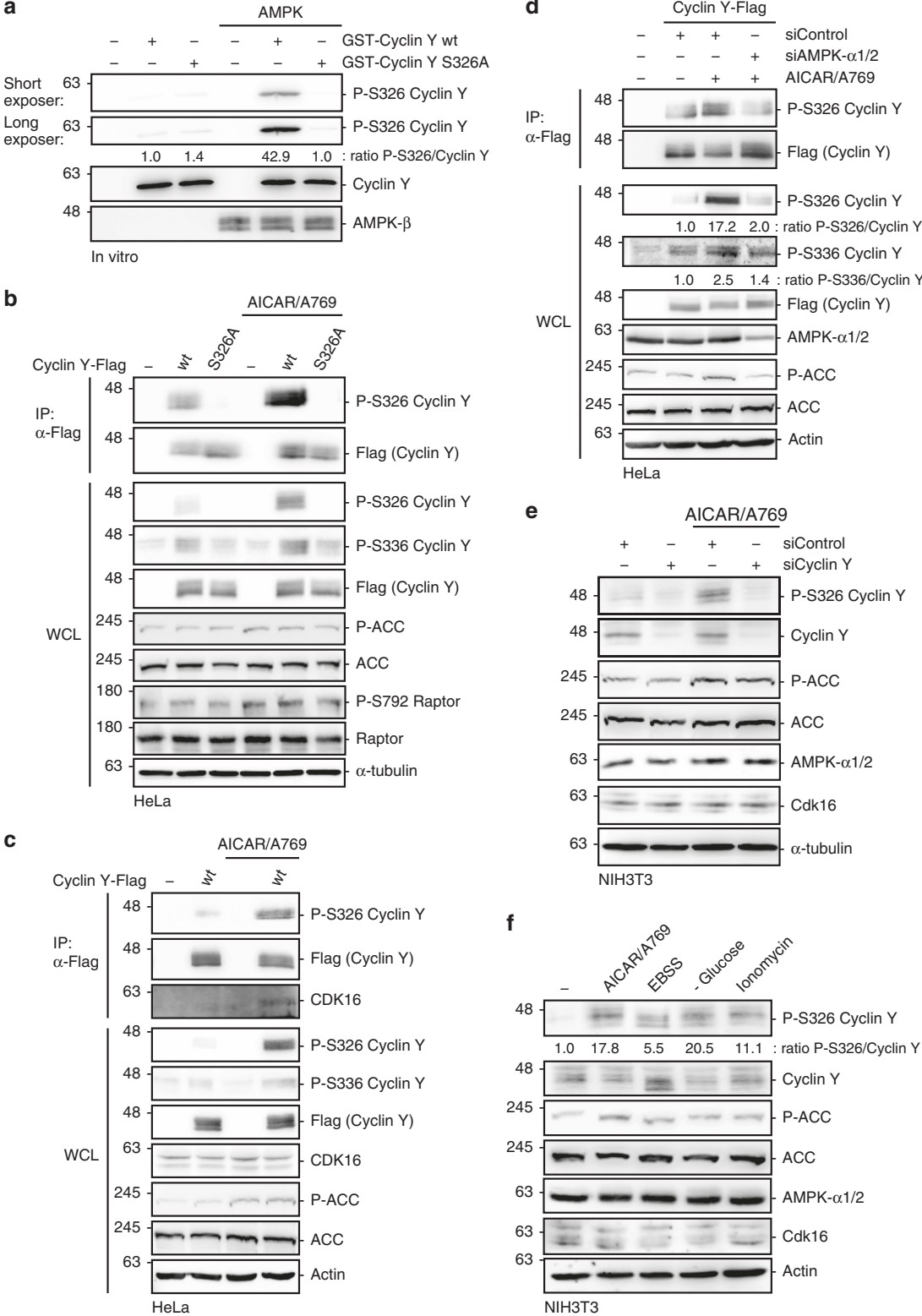

S336. Previously, we identified CDK16-dependent S336 phosphorylation of Cyclin Y as cellular readout of CDK16 kinase activity[15,22]. The CDK16-dependent phosphorylation at S336 was inhibited in the Cyclin Y-S326A mutant (Fig. 2b), suggesting that S326 phosphorylation influences the interaction with CDK16. Indeed, after activating AMPK with AICAR/A769662, we observed enhanced interaction between Cyclin Y-Flag and endogenous CDK16 (Fig. 2c). Cyclin Y/CDK16 complex formation was accompanied by an increase in CDK16 kinase activity as measured by S336 phosphorylation of Cyclin Y (Fig. 2c). Furthermore, we investigated the need for AMPK to activate Cyclin Y/CDK16. Knockdown of the AMPK-α1 and α2 subunits diminished Cyclin Y S326 phosphorylation in response to AICAR/A769662 (Fig. 2d).

In addition, we validated the AMPK-dependent phosphorylation of Cyclin Y at S326 in the LKB1-positive U2OS cell line.

**Fig. 2 Verification of the Cyclin Y S326 phosphorylation site in vitro and in cells. a** GST-Cyclin Y wt and S326A were incubated ± AMPK. S326-P was detected using a phospho-specific antibody ($n = 2$). **b** HeLa cells were transfected with vectors expressing Cyclin Y-Flag, the S326A mutant or an empty control vector and treated with 0.5 mM AICAR/50 μM A769662 (A769) for 1 h as indicated. Cyclin Y was immunoprecipitated with a Flag antibody (IP). The blots were immunoblotted with the indicated antibodies (WCL). ($n = 2$). **c** HeLa cells were transfected with a vector expressing Cyclin Y-Flag as indicated. After 72 h cells were treated with 0.5 mM AICAR/50 μM A769662 for 1 h. Cyclin Y was immunoprecipitated with a Flag antibody. The blots were immunoblotted with the indicated antibodies ($n = 2$). **d** HeLa cells were transfected with a vector expressing Cyclin Y-Flag and siRNA against AMPK-α1/2 or control siRNA as indicated. After 72 h cells were treated with 0.5 mM AICAR/50 μM A769662 for 1 h. The blots were immunoblotted with the indicated antibodies ($n = 2$). **e** NIH3T3 cells were transfected with the indicated siRNAs. After 72 h cells were treated with 0.5 mM AICAR/50 μM A769662 for 1 h prior to protein analysis by immunoblotting ($n = 2$). **f** NIH3T3 cells were treated with 0.5 mM AICAR/50 μM A769662, EBSS or Ionomycin for 1 h or were grown in glucose-free medium for 16 h and lysates were immunoblotted as indicated ($n = 2$). n biological independent replicate. Source data are provided as a Source Data file.

AICAR/A769662 treatment of U2OS cells enhanced S326 phosphorylation of Cyclin Y (Supplementary Fig. 2c). In parallel, the interaction between Cyclin Y-Flag and endogenous CDK16 was stimulated (Supplementary Fig. 2c), and its kinase activity increased as measured by phosphorylation of Cyclin Y at S336. Similar as in HeLa cells, the AMPK-α1 and α2 subunits were necessary for Cyclin Y S326 phosphorylation in response to AICAR/A769662 (Supplementary Fig. 2d). The phosphorylation of Cyclin Y by AMPK was supported in U2OS cells by the interaction between AMPK and Cyclin Y. Using co-immunoprecipitation, we observed a weak interaction between the endogenous AMPK-α1/2 and Cyclin Y S326A in response to AICAR/A769662 (Supplementary Fig. 3a). With the help of proximity ligation assays the interaction between endougens Cyclin Y and AMPK-β1/2 was measured, which slightly but significantly increased after treating cells with AICAR/A769662 or EBSS (Supplementary Fig. 3b, c). The specificity of the Cyclin Y antibody 2C9E3 was determined for immunoblotting and immunostaining with the help of siRNA-mediated knockdown of endogenous Cyclin Y in U2OS cells (Supplementary Fig. 3d, e).

Phosphorylation of endogenous Cyclin Y at S326 was tested in NIH3T3 cells. Upon AICAR/A769662 treatment S326 phosphorylation increased, with the specificity of phosphorylation verified upon knockdown of Cyclin Y (Fig. 2e). Similar to the AICAR/A769662 stimulation, S326 phosphorylation was also enhanced in NIH3T3 cells in response to glucose or amino acid (EBSS) starvation, as well as upon increased $Ca^{2+}$ signaling (ionomycin), i.e., conditions known to stimulate AMPK activity (Fig. 2f). Collectively, these results provide evidence for AMPK-dependent phosphorylation of Cyclin Y at S326, which promotes Cyclin Y/CDK16 interaction and catalytic activity of CDK16.

**Cyclin Y and CDK16 are required for induction of autophagy.** AMPK is an important regulator of autophagy[6]. Since phosphorylation of Cyclin Y at S326 was stimulated in response to autophagy-inducing conditions (Fig. 2f), we hypothesized that Cyclin Y/CDK16 might affect autophagy. In support of this idea are the involvement of Cyclin Y/CDK16 in vesicle transport and membrane fusion processes[23–25], both hallmarks of autophagy[26]. We first addressed the role of Cyclin Y or Cdk16 in response to amino acid starvation in NIH3T3 cells. The formation of autophagy-associated LC3-II and the degradation of the autophagic adaptor protein p62 were inhibited upon knockdown of Cdk16 or Cyclin Y (Fig. 3a). In addition, autophagy-associated LC3 puncta formation was reduced to basal level upon knockdown of Cyclin Y or Cdk16 (Fig. 3b, c). These findings provided support for a role of Cyclin Y and Cdk16 in autophagy. Of interest is the missing induction of S326 phosphorylation in the absence of CDK16.

We investigated the autophagic flux by combining Bafilomycin A1 (Baf. A1) treatment with EBSS-mediated starvation. LC3-II

formation upon amino acid starvation was further increased by Baf. A1. Cyclin Y knockdown inhibited the LC3 lipidation, suggesting that Cyclin Y is required for the EBSS-induced autophagic flux (Fig. 3d). For control, 3-methyladenine (3-MA) was applied, which blocks the activity of the VPS34 PI3K complex and thus inhibits an early step in autophagy[27]. Next, we addressed the role of Cdk16 using Cdk16$^{+/+}$ and Cdk16$^{-/-}$ mouse embryo fibroblasts (MEFs) by treating the cells with EBSS and Baf. A1. In the absence of Cdk16 the lipidation and the puncta formation of LC3 was substantially reduced in immortalized cells (Fig. 3e–g).

We also studied the autophagy marker WIPI1 and Wipi2[28]. Puncta formation of endogenous Wipi2b was reduced in Cdk16$^{-/-}$ MEFs compared to Cdk16$^{+/+}$ MEFs in response to starvation (Supplementary Fig. 4a, b). Similarly, in cells stably expressing GFP-WIPI1 we observed reduced puncta formation in Cdk16$^{-/-}$ MEFs compared to Cdk16$^{+/+}$ MEFs in response to starvation (Supplementary Fig. 4c, d). To evaluate whether the autophagic defect in the Cdk16$^{-/-}$ MEFs resulted from the depletion of Cdk16, we introduced human CDK16 (human splice isoform 1 (Uniprot Q00536-1)) into these cells (Fig. 3h). Its protein expression level was comparable to that of the endogenous Cdk16. Whether the two protein bands detected for the endogenous mouse Cdk16 correspond to different splice isoforms or are the result of posttranslational modifications is not known (Fig. 3h). Importantly, human CDK16 wild-type (wt) rescued the autophagy defect upon EBSS treatment in reconstituted Cdk16$^{-/-}$ MEFs (Fig. 3h). Together, these data suggest that Cyclin Y and Cdk16 regulate autophagy.

**Autophagy induction needs the active Cyclin Y/CDK16 complex.** In the following, we addressed whether the interaction of Cyclin Y and CDK16 and the kinase activity of the complex were important to control autophagy. We employed the binding deficient Cyclin Y-L222A/S224A mutant (Cyclin Y-AA) and the kinase-deficient CDK16 K194R mutant (CDK16-KR) mutant[14]. The inability of Cyclin Y-AA to interact with and activate CDK16 was verified (Supplementary Fig. 5a). While over-expression of the Cyclin Y/CDK16 complex in NIH3T3 cells induced LC3 lipidation to an extent comparable to EBSS treatment, Cyclin Y/CDK16-KR or Cyclin Y-AA/CDK16 failed to stimulate LC3 lipidation (Fig. 4a). This was further verified using NIH3T3 cells stably expressing mCherry-GFP-LC3[29]. This allows distinguishing autophagosomes from autolysosomes, because the red, acid insensitive mCherry can be measured in both compartments while the green, acid sensitive GFP can only be detected in the former. Thus, autophagosomes appear yellow (a combination of red and green) and autolysosomes red, permitting the analysis of autophagic flux. The expression of the Cyclin Y/CDK16 complex increased the puncta formation of yellow and red dots similarly to EBSS treatment, while Baf. A1 promoted the accumulation of yellow autophagosomes (Fig. 4b, c). A significantly lower number of autophagosomes and autolysosomes were measured upon

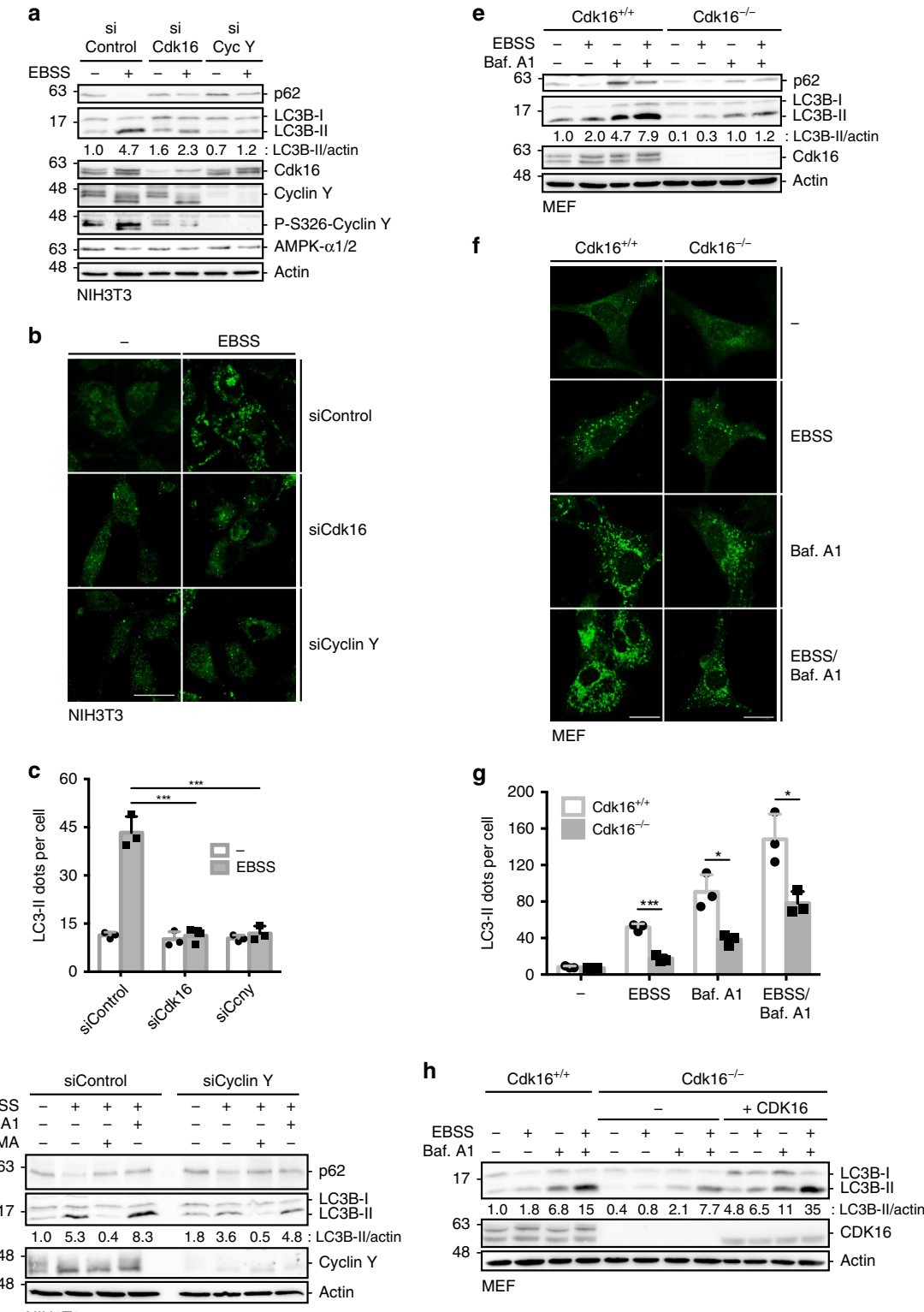

expression of Cyclin Y/CDK16-KR or Cyclin Y-AA/CDK16. A similar effect was observed by reconstituting the Cdk16$^{-/-}$ MEFs with a GFP-CDK16-KR mutant in comparison to CDK16-wt (Supplementary Fig. 5b). These results suggest that active Cyclin Y/CDK16 accelerates autophagic flux.

Furthermore, we determined the ability of Cyclin Y/CDK16 to induce autophagy in HeLa cells. As for NIH3T3 cells, LC3-II formation and LC3 puncta increased in response to Cyclin Y/CDK16 expression due to enhanced autophagic flux (Supplementary Fig. 6a–c). Flux inhibition by the lysosomal protease inhibitors Pepstatin A (PepA) and E64, or by Baf. A1, led to an additional increase of LC3-II and LC3 puncta (Supplementary Fig. 6d–f). 3-MA abrogated the ability of Cyclin Y/CDK16 to stimulate autophagy (Supplementary Fig. 6d–f). Also, Cyclin Y/CDK16-KR and Cyclin Y-AA/CDK16 were less efficient in inducing autophagy (Supplementary Fig. 6a–c). These results confirm the autophagy-promoting activity of Cyclin Y/CDK16 complexes in human cells.

**Fig. 3 Cyclin Y and CDK16 are required for efficient induction of autophagy. a** NIH3T3 cells were transfected with siRNA against Cdk16, Cyclin Y or a control for 72 h and grown in EBSS for 2 h. Proteins were detected as indicated (*n* = 3). **b** Representative confocal images of the NIH3T3 treated as in panel **a**. Endogenous LC3 was stained with the 4E12 antibody to monitor autophagy. Scale bar: 50 µm. **c** Quantification of the LC3 dots shown in panel **b**. Statistical significance was measured via unpaired and two-tailed Student's *t*-tests and is presented as follows: ***$p$ < 0.001. All error bars indicate SD (*n* = 3; 100 cells counted for each replicate; siControl vs. siCdk16: t = 10.09, df = 4; siControl vs. siCyclin Y: t = 9.866, df = 4). **d** NIH3T3 cells were transfected with siControl or siCyclin Y, treated with 200 nM Bafilomycin A1 (Baf. A1) or 3-MA for 2 h, and grown in EBSS for additional 2 h as indicated. Lysates were immunoblotted with the indicated antibodies (*n* = 2). **e** Immortalized Cdk16$^{+/+}$ and Cdk16$^{-/-}$ MEFs were treated with 200 nM Baf. A1 for 4 h prior to growth in EBSS for additional 2 h. Proteins were measured by immunoblotting (*n* = 3). **f** Representative confocal images of the Cdk16$^{+/+}$ and Cdk16$^{-/-}$ MEFs treated as in panel **e**. Endogenous LC3 was stained (antibody 4E12) to monitor autophagy. Scale bar: 50 µm. **g** Quantification of the LC3 dots shown in panel **f**. Statistical significance was measured via unpaired and two-tailed Student's *t*-tests and is presented as follows: *$p$ < 0.05 and ***$p$ < 0.001. All error bars indicate SD (*n* = 3; 100 cells counted for each replicate; EBSS: t = 10.850, df = 4; Baf. A1: t = 4.58, df = 4; EBSS/Baf. A1: t = 3.978, df = 4). **h** Human CDK16 was stably expressed in immortalized Cdk16$^{-/-}$ MEFs ( + CDK16). Control cells were infected with an empty virus. Cells were treated with 200 nM Baf. A1 and grown in EBSS. Proteins were detected as indicated (*n* = 2). *n* biological independent replicate. SD standard deviation. Source data are provided as a Source Data file.

**CDK14 and CDK15 show no involvement in autophagy**. Besides CDK16, Cyclin Y also interacts with and activates CDK14[30,31]. Therefore, we addressed whether Cyclin Y/CDK14 was able to regulate autophagy. Because of its high homology to CDK14, we also tested CDK15, the second member of the PFTAIRE family. Cyclin Y interacted with all three kinases, which were active as monitored by S336 phosphorylation of Cyclin Y (Fig. 4d). However, when LC3 lipidation and puncta formation were measured, only CDK16 was capable of stimulating autophagy (Fig. 4d–f), indicating selectivity between the three Cyclin Y-kinase complexes.

**Cyclin Y/CDK16-induced autophagy requires ULK1 and Beclin1**. Several autophagy-related pathways use LC3 lipidation, including LC3-associated phagocytosis (LAP) and the LC3 conjugation system for Interferon-γ-mediated pathogen control[26]. However, these pathways are independent of ULK1[32]. In addition, ATG5-dependent and Beclin1-independent cases of autophagy have been described[32]. To determine whether Cyclin Y/CDK16 promotes autophagy dependent on ULK1 and/or Beclin1, we interfered with ULK1 and Beclin1 expression. The knockdown of ULK1 or Beclin1 inhibited the Cyclin Y/CDK16-dependent autophagy (Fig. 5a–c). Therefore, activation of autophagy by Cyclin Y/CDK16 appears to require both the ULK1 and Beclin1 complexes.

**Cyclin Y/CDK16 is a downstream effector of AMPK**. Our findings suggest that AMPK regulates autophagy through Cyclin Y/CDK16. This was addressed directly by activating AMPK in Cdk16$^{-/-}$ MEFs and monitoring autophagy. While ACC was phosphorylated independently of Cdk16 (Fig. 6a), LC3 conversion and puncta formation as well as the autophagic flux were reduced in the absence of Cdk16 (Fig. 6a–d). Comparable findings were made in NIH3T3 cells where Cyclin Y knockdown decreased autophagy (Fig. 6e–g). These results provide support for Cyclin Y/Cdk16 acting as a downstream effector of AMPK during the induction of autophagy.

Next, we addressed the role of Cyclin Y S326 phosphorylation. Compared to wild-type Cyclin Y, the S326A mutant was unable to induce autophagy in HeLa cells (Fig. 7a–c), probably because this mutant cannot interact with and activate CDK16 (Figs. 2b and 7a). In support of this interpretation, Cyclin Y S100A, a CDK16 binding deficient mutant[15], did neither co-immunoprecipitate with CDK16 nor induce autophagy (Fig. 7a–c). In contrast, mutation of S324A, which is neighboring the S326 phosphorylation site, had no effect (Fig. 7a–c and Supplementary Fig. 7a). To expand on the relevance of the S326 phosphorylation site, we tested whether the S326E and S326D

mutants were capable of mimicking S326 phosphorylation. However, both mutants bound with reduced affinity to CDK16 and, accordingly, induced LC3 lipidation less efficiently (Supplementary Fig. 7b). Thus, an acidic residue was insufficient to mimic S326 phosphorylation. Notably, the phosphorylation of S324 has been described in multiple proteome analyses (www.phosphosite.org/uniprotAccAction?id=Q8ND76)[33], which might be affected by S326E and S326D. However, the binding of the double mutants S324A/S326D and S324A/S326E to CDK16 was even weaker (Supplementary Fig. 7b). A reason for the failure of the phospho-mimic mutants S326E and S326D to induce autophagy might be their inability to recruit 14-3-3 proteins, which are required for activation of CDK16. Indeed, the phospho-mimic mutants behaved like the S100A and S326A mutants, which did not interact with 14-3-3 (Supplementary Fig. 7a, b)[15]. Thus, the data suggest that phosphorylation of Cyclin Y at S326 stimulates the interaction with CDK16 and activates CDK16 by recruiting 14-3-3 proteins.

To understand whether the activity of AMPK is necessary for the Cyclin Y/CDK16-mediated induction of autophagy, we suppressed AMPK activity by knockdown of the α1 and α2 subunits or by treating cells with the AMPK inhibitor Compound C. Under both conditions Cyclin Y/CDK16-dependent autophagy was impaired (Fig. 7d–f and Supplementary Fig. 7c, d). Thus, AMPK is required for phosphorylation of S326 to promote Cyclin Y/CDK16 interaction, activation of CDK16 catalytic activity and autophagy.

## Discussion

We identified Cyclin Y/CDK16 as AMPK substrate and as autophagy effector. Cyclin Y is phosphorylated at S326 by AMPK in vitro and in cells, which promotes interaction with its kinase partner CDK16 by recruitment of 14-3-3 proteins leading to activation of the catalytic activity of this complex. The functional analysis revealed that Cyclin Y/CDK16 induces autophagy, dependent on S326 phosphorylation and requires ULK1 and Beclin1, which are both necessary for the induction of macro-autophagy[32]. In the absence of functional Cyclin Y/CDK16, the induction of autophagy by AMPK activating stimuli was hampered, indicating that this kinase is an important signaling component downstream of AMPK. Together, our findings provide evidence for active Cyclin Y/CDK16 being a AMPK effector in autophagy (Fig. 8).

We performed kinase assays on protein microarrays to extend the understanding of the cellular functions of AMPK. Our screen revealed 63 potential AMPK substrates (Fig. 1 and Supplementary Data 1). In comparison to previously published studies[34–36], the use of protein microarrays allows substrate identification independently of cell type-specific protein expression and of

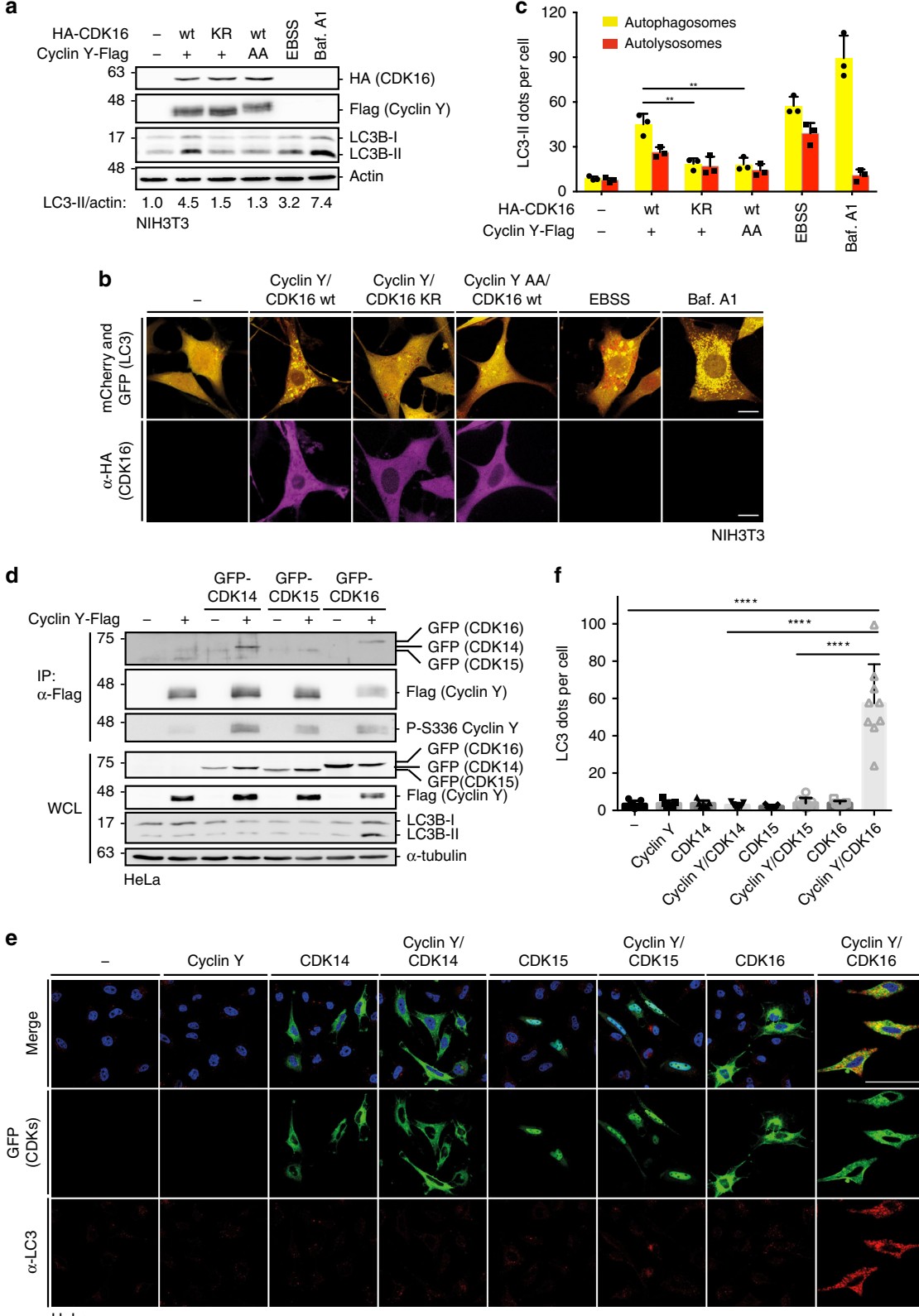

antibodies recognizing a phosphorylated AMPK consensus sequence. However, coverage of the proteome is limited to proteins available on the microarray. Furthermore, the in vitro identified substrates require verification in cells. This is critical and is illustrated by our findings regarding CDK16 phosphorylation as all the in vitro AMPK sites on CDK16 could not be confirmed as AMPK targets in cells. Thus, substoichiometric phosphorylation of several sites within one target protein may result in false positives. In contrast, the single site at S326 identified on Cyclin Y was confirmed in cells and the extent of phosphorylation paralleled AMPK activity (Figs. 1 and 2).

**Fig. 4 Active Cyclin Y/CDK16 complexes induce autophagy. a** NIH3T3 cells stably expressing mCherry-GFP-LC3 were transfected with HA-CDK16 and Cyclin Y-Flag as indicated or treated for 2 h with EBSS or for 4 h with 200 nM Bafilomycin A1 (Baf. A1) and lysates were immunoblotted as indicated. KR kinase-deficient CDK16 mutant, AA CDK16 binding deficient Cyclin Y mutant ($n = 3$). **b** Representative confocal images of the NIH3T3-mCherry-GFP-LC3 cells treated as in panel **a**. Staining of the HA-CDK16 in purple identified transfected cells. Autophagosomes (yellow dots) and autolysosomes (red dots) were detected by an overlay of the GFP and mCherry fluorescent signals. Scale bar: 20 µm. **c** Quantification of autophagosomes (yellow dots) and autolysosomes (red dots) of cells shown in panel **b**. Statistical significance was measured via unpaired and two-tailed Student's $t$-tests and is presented as follows: \*\*$p < 0.01$. All error bars indicate SD ($n = 3$; 50 cells counted for each replicate; wt vs. KR: $t = 5.707$, df $= 4$; wt vs. AA: $t = 5.557$, df $= 4$). **d** GFP-tagged CDK14, CDK15 or CDK16 were expressed with or without Cyclin Y-Flag in HeLa cells. Cyclin Y was immunoprecipitated with a Flag antibody (IP). Lysates were immunoblotted with the indicated antibodies (WCL). ($n = 4$) **e** Representative confocal images of HeLa cells treated as in panel **d**. Fluorescent GFP signals identified CDK expressing cells. Endogenous LC3 (red) was used to measure autophagy with the 4E12 antibody. Scale bar: 50 µm. **f** Quantification of the LC3 dots shown in panel **e**. Statistical significance was measured via unpaired and two-tailed Student's $t$-tests and is presented as follows: \*\*\*\*$p < 0.0001$. All error bars indicate SD. ($n = 1$; 100 cells were counted for each treatment; control vs. Cyclin Y/CDK16: $t = 6.771$, df $= 14$; Cyclin Y/CDK14 vs. Cyclin Y/CDK16: $t = 6.855$, df $= 14$; Cyclin Y/CDK15 vs. Cyclin Y/CDK16: $t = 7.139$, df $= 15$). n biological independent replicate. SD standard deviation. Source data are provided as a Source Data file.

Both CDK16 and CDK14 were reported to interact with Cyclin Y[14,31]. We found that CDK15 also uses Cyclin Y as regulatory subunit (Fig. 4d). Since Cyclin Y is required for AMPK-stimulated autophagy (Fig. 6), we addressed the functionality of all three complexes. However, neither Cyclin Y/CDK14 nor Cyclin Y/CDK15 induced autophagy (Fig. 4d–f). Likely, this is because the three complexes possess unique substrate specificities[13]. Little is known regarding Cyclin Y/CDK14 and Cyclin Y/CDK15 substrates. Cyclin Y/CDK14 participates in the activation of the Wnt signaling pathway particularly in the $G_2/M$-phase, consistent with differential expression pattern of CDK14 during the cell cycle, while CDK16 is present throughout the cell cycle[30,37]. Interestingly, the interaction of CDK14 with Cyclin Y has also been proposed to depend on S326 phosphorylation[38]. For CDK15 no functional data are available. Together, the available evidence supports the hypothesis that substrate specificity is different of these three Cyclin Y kinase complexes.

Autophagy is one of the processes that AMPK stimulates under starvation conditions to provide essential building blocks to the cell[39]. AMPK phosphorylates and controls important upstream regulators of the autophagic cascade, including the ULK1 and the VPS34-dependent PI3K complexes that govern initial steps in the formation of phagophores giving rise to autophagosomes[9–11]. However, the molecular mechanisms of autophagy regulation are not fully resolved and may not be linear[40]. With Cyclin Y/CDK16 we have identified an additional effector of AMPK. A potential link to autophagy is further suggested by the function of CDK16 in vesicular transport[25] and in actin cytoskeleton organization[41–43], as both these processes are relevant for autophagy[44–46]. For example, the ER-Golgi intermediate compartment has been suggested as a membrane source for autophagosome biogenesis, a process that is upstream and independent of ULK1 and PI3K complexes[47]. Our data suggest that AMPK activates Cyclin Y/CDK16 to initiate an early step in the formation of autophagosomes. These findings are consistent with the published observations and it will now be interesting to identify and define the functions of Cyclin Y/CDK16 substrates.

With the association of autophagy with disease processes, it is notable that the CCNY locus, encoding Cyclin Y, has been associated with inflammatory bowel disease (IBD)[48–50]. This is interesting in light of the observation that defective autophagy is strongly linked to IBD pathogenesis[51]. The phenotype of Cdk16$^{-/-}$ and Cyclin Y-like$^{-/-}$ mice during spermatogenesis also points to a function in autophagy. Cyclin-Y-like is the homologue of Cyclin Y predominantly expressed in testis and mutants of Cyclin Y-like show sperm maturation defects similar to Cdk16. Spermatozoa of both mutants display thinning of the annulus region, malformed heads and excess residual cytoplasm, which leads to an impaired motility[14,52,53]. Similar phenotypes

were observed in Sertoli cell specific Atg5$^{-/-}$ and germ cell specific Atg7$^{-/-}$ mice[54–56]. A reason for these differentiation defects seems to be a failure to reorganize the cytoskeleton during the final differentiation step of the spermatozoa[55,56]. For CDK16 several studies described a significant upregulation of this kinase in different tumor entities compared to adjacent normal tissue. The elevated expression of CDK16 correlates positively with the tumor aggressiveness and is required for tumor cell proliferation[57–61]. In light of these recent findings, our data suggest that CDK16 promotes tumor growth by enhancing tumor-supportive autophagy[62]. Collectively, CDK16 emerges as a potential drug target.

## Methods

**Material**. Material used are listed in Supplementary Table 3.

**Contact for reagent and resource sharing**. Further information and requests for resources and reagents should be directed to and will be fulfilled by Jörg Vervoorts (jvervoorts-weber@ukaachen.de).

**Experimental model and subject details**. A detailed step-by-step protocol how we stimulate AMPK in cells and measure autophagy is available in Methods in Molecular Biology[63].

**Cell lines and cell culture**. HeLa (ATCC, CCL-2), U2OS (ATCC, HTB-96), COS7 (ATCC, CRL-1651), NIH3T3 (ATCC, CRL-1658), Cdk16$^{-/-}$, and Cdk16$^{+/+}$ MEFs (provided by Stephan Geley[14]) were cultured in Dulbecco's modified Eagle Medium (DMEM) (Gibco, Thermo Fisher Scientific) containing 10% heat-inactivated fetal calf serum (FCS) (Gibco, Thermo Fisher Scientific) and 1% Penicillin/Streptomycin (Gibco, Thermo Fisher Scientific). NIH3T3 cells stably expressing mCherry-GFP-LC3 were generated by viral infection of pBABE-puro-mCherry-EGFP-LC3B[64,65] and cultivated in medium supplemented with 3 µg/ml Puromycin (Sigma-Aldrich). Cdk16$^{-/-}$ and Cdk16$^{+/+}$ MEFs were immortalized by viral infection with the pRetro-Super-blasti-shRNA-p19$^{ARF}$ construct, which prevents activation of p53 by promoting Mdm2 activity[66].The immortalized MEFs were grown in medium containing 20 µg/ml Blasticidin (InvivoGen). Immortalized Cdk16$^{-/-}$ MEFs stably expressing GFP-WIPI1 were generated by retroviral infection of pMXs-IP GFP-WIPI-1[67]. Reconstitution of immortalized Cdk16$^{-/-}$ MEFs with human CDK16 wild-type was performed by retroviral infection of pBABE-puro hCDK16-wt. Reconstitution of immortalized Cdk16$^{-/-}$ MEFs with GFP-CDK16-wt or the kinase-deficient GFP-CDK16-K194R/D304N mutant were performed by lentiviral infection with the Doxycycline-inducible pLIX403-GFP-CDK16-wt or pLIX403-GFP-CDK16-K194R/D304 constructs. In order to obtain a comparable expression to the CDK16$^{+/+}$ MEFs the expression of GFP-CDK16-wt and GFP-CDK16-K194R/D304N were induced 16 h before treating the cells with EBSS by 200 ng Doxycycline/ml and 100 ng Doxycycline/ml respectively. All viral infected cells were cultured in full medium supplemented with 2 µg/ml Puromycin and 20 µg/ml Blasticidin.

All cells were maintained at 37 °C in a humidified incubator in an atmosphere of 5% $CO_2$ and passaged with trypsin every 1–3 days when the cells reached 80% confluency. Sf9 insect cells (ATCC, CRL-1711) were cultured in Grace's Medium (Gibco, Thermo Fisher Scientific) supplemented with 10% heat-inactivated FCS and 20 µg/ml Gentamycin (Merck Millipore) at 27 °C.

Cells were regularly tested for contamination with Mycoplasma using a PCR-based detection analysis and discarded if tested positive. The HeLa, NIH3T3 and

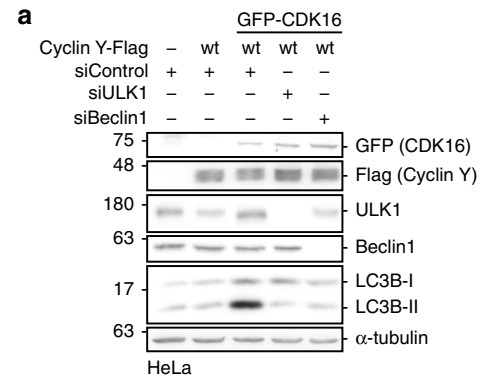

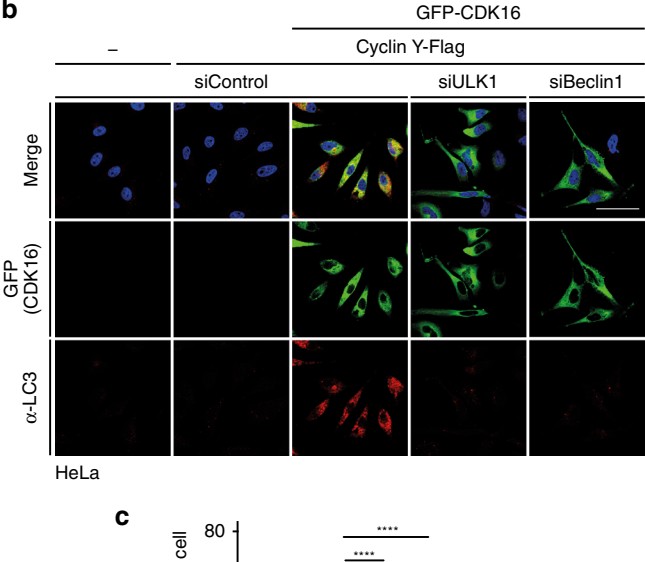

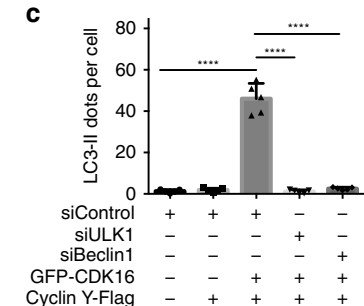

**Fig. 5 ULK1 and Beclin1 are required for Cyclin Y/CDK16-induced autophagy. a** HeLa cells were co-transfected with siRNA against ULK1, Beclin1 or a control and vectors expressing GFP-CDK16 and Cyclin Y-Flag for 72 h. Lysates were immunoblotted with the indicated antibodies. ($n = 3$). **b** Representative confocal images of HeLa cells treated as in panel **a**. GFP-CDK16 identified transfected cells. Endogenous LC3 staining (red) with the 4E12 antibody monitored autophagy. Scale bar: 50 μm. **c** Quantification of the LC3 dots shown in panel **b**. Statistical significance was measured via unpaired and two-tailed Student's $t$-tests and is presented as follows: ****$p < 0.0001$. All error bars indicate SD. ($n = 1$; 250 cells were analyzed for each treatment; siControl vs. siControl Cyclin Y/CDK16: $t = 13.49$, df $= 8$; siControl Cyclin Y/CDK16 vs. siULK1 Cyclin Y/CDK16: $t = 13.53$, df $= 8$; siControl Cyclin Y/CDK16 vs. siBeclin1 Cyclin Y/CDK16: $t = 13.15$, df $= 8$). n biological independent replicate. SD standard deviation. Source data are provided as a Source Data file.

U2OS cells have been authenticated by PCR-single-locus-technology using the PowerPlex 21 PCR kit (Promega) carried out by Eurofins Medigenomix Forensik GmbH (Certificates are available upon request).

**Stimulation of cell lines**. To induce autophagy, the medium of the cells was replaced by Earle's balanced salt solution (EBSS) (Gibco, Thermo Fisher Scientific)

for 2 h or by DMEM without glucose (Gibco, Thermo Fisher Scientific) with heat-inactivated 10% FCS (Gibco, Thermo Fisher Scientific) for 16 h. To investigate the induction of autophagy in viral infected cells antibiotics for selection were omitted one week before the experiment. Where indicated, cells were treated with the following compounds: 2 mM 3-Methyladenine (3-MA; Merck Millipore), 50 μM A-769662 (A769; InvivoGen), 0.5 mM AICAR (Tocris Bioscience), 1 μM Ionomycin (Sigma-Aldrich), 200 nM Bafilomycin A1 (Baf. A1; Enzo Life Sciences), 40 μM Compound C (Merck Millipore), 10 μg/ml E64-d (Enzo Life Sciences), 10 μg/ml Pepstatin A (AppliChem).

**Transient transfections of cell lines**. Transfections of HeLa or U2OS cells with plasmids or mixtures of plasmids and siRNA were performed by the calcium phosphate (CaPO₄) method. Briefly, HeLa or U2OS cells were seeded at a density of $1 \times 10^6$ cells per 10 cm culture dish. On the following day, a total of 20 μg plasmid DNA (optional: plus 50 nM siRNA oligos) was diluted with 950 μl 1 × HBS buffer (21 mM HEPES (pH 6.95), 137 mM NaCl, 5 mM KCl, 0.71 mM Na₂HPO) and mixed by vortexing. The Cyclin Y and CDKs expressing constructs were transfected each with 4 μg plasmids DNA and filled up to 20 μg with empty plasmid. Afterwards 50 μl of 2.5 M CaCl₂ were added to the solution dropwise under constant shaking and subsequently mixed by vortexing. This mixture was incubated at room temperature for 30 min to allow the formation of CaPO₄-DNA precipitates. Thereupon the transfection solution was added to the cell culture dish dropwise under gentle rocking. The cells were incubated at 37 °C for 16–24 h, afterwards washed gently with pre-warmed HEPES (10 mM HEPES (pH 7.3), 142 mM NaCl, 6.7 mM KCl) and incubated at room temperature for 10 min prior to addition of fresh medium. When only protein-expressing plasmids were used, cells were harvested 48 h after the beginning of transfection. Experiments with RNAi were harvested 72 h after transfection.

NIH3T3 cells were transfected with plasmids by Lipofectamine 2000. Briefly, NIH3T3 cells were seeded at a density of $4 \times 10^5$ cells per 6 cm culture dish. On the following day, a total of 8 μg plasmid DNA was diluted with 250 μl OptiMEM and mixed gently. For each transfection sample 9 μl Lipofectamine 2000 (ThermoFisher) were diluted with 250 μl OptiMEM separately. After 5 min incubation, the diluted DNA was combined with the diluted Lipofectamine 2000, mixed gently and incubated for 20 min at room temperature. Afterwards the transfection mix was added to the cell culture dish dropwise under gentle rocking. The cells were incubated at 37 °C and fresh medium was changed after 24 h, in total cells were harvested 48 h after transfection.

For the RNAi experiments with NIH3T3, cells were transfected with the indicated siRNA oligo pools listed in the Supplementary Table 3 using HiPerFect (Qiagen). Therefore cells were seeded onto 6-well culture dishes at a density of $5 \times 10^5$ cells per well. After 1 h, 5 μl of a 20 μM dilution of the according siRNA were mixed with 400 μl OptiMEM and 15 μl HiPerFect transfection reagent. After incubation at room temperature for 10 min the transfection mixture was added dropwise to the cells under gentle rocking leading to a final siRNA concentration of 50 nM in the culture medium. Afterwards the cells were incubated under usual conditions for 72–96 h to allow the siRNA-mediated knockdown of proteins. For transfections the cells were seeded without antibiotics.

In smaller culture dishes the number of cells was scaled down according to the growth area of the culture dishes while the volumes of the buffers and solutions were adjusted in relation to the volume of culture medium.

**Viral infection of MEFs**. Retroviruses were generated using the pVPack system from Agilent Technologies consisting of the pVPack-GP and pVPack-Eco vectors together with pBABE-puro, pBABE-puro-hCDK16-wt, or pMXs-IP-GFP-WIPI1. Lentiviruses were generated using pMDLg/pRRE, pRSV-REV, and pCMV-VSV-G vectors together with pLIX403-GFP-CDK16-wt or pLIX403-GFP-CDK16-K194R/D304N. COS7 cells were transiently transfected via the calcium phosphate method with the viral constructs together with the envelope expressing vector and the gag-pol element expression vector for 6 h, afterwards the medium was changed to DMEM containing 10% FCS. After 24 h the virus-containing supernatant of these cultures was collected and filtered through a 0.45 μm PVDF filter (Merck Millipore). MEFs cells were incubated with the filtered supernatant supplemented with 4 μg/ml Polybrene (Sigma-Aldrich) overnight. The next day the MEFs cells were replated in fresh medium and a second round of infection with fresh supernatant was performed after 24 h to maximize the rate of infection. The Puromycin selection (2 μg/ml, Sigma-Aldrich) started 48 h after the last infection and was continued for 2 weeks to generate stable cell pools.

**Cloning and mutagenesis**. A cDNA clone for CCNYiso1 was purchased from Source BioScience (clone ID: IRAKp961O14141Q) and separately cloned into pGEX-4T1, pVL1392, and pcDNA3-FLAG using restriction digestion. CCNY and CDK16 mutation constructs were generated with QuikChange II Site-Directed Mutagenesis Kit II (Agilent Technologies), the used oligonucleotides are listed in the Supplementary Table 3. The cDNA for CDK14 (clone ID: IRCBp5005L0611Q) and CDK15 (clone ID: IRATp970G0369D) were purchased from Source BioScience and subcloned into pEGFP-C3.

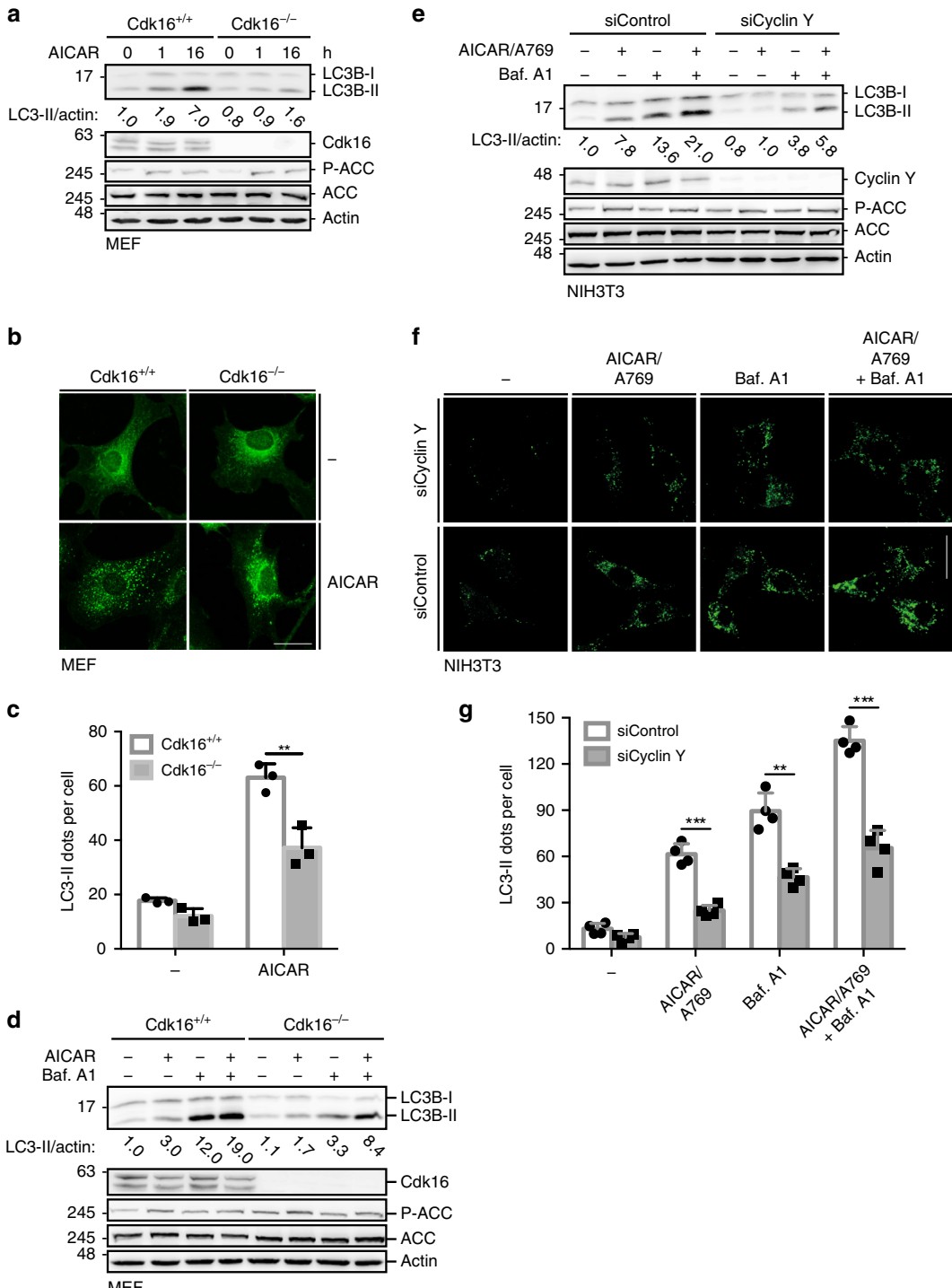

**Fig. 6 AMPK-induced autophagy requires Cyclin Y/CDK16. a** Immortalized Cdk16$^{+/+}$ and Cdk16$^{-/-}$ MEFs were treated with 1 mM AICAR for 1 or 16 h. Lysates were immunoblotted with the indicated antibodies ($n = 3$). **b** Representative confocal images of the Cdk16$^{+/+}$ and Cdk16$^{-/-}$ MEFs treated as in panel **a**. Endogenous LC3 Staining with the 4E12 antibody monitored autophagy. Scale bar: 50 μm. **c** LC3 dots of experiments as shown in panel **b** were quantified. Statistical significance was measured via unpaired and two-tailed Student's $t$-tests and is presented as follows: **$p < 0.01$. All error bars indicate SD. ($n = 3$; 100 cells counted for each replicate; Cdk16$^{+/+}$ + AICAR vs. Cdk16$^{-/-}$ + AICAR: t = 4.964; df = 4). **d** Immortalized Cdk16$^{+/+}$ and Cdk16$^{-/-}$ MEFs were treated with 1 mM AICAR for 1 h and/or with 200 nM Bafilomycin A1 (Baf. A1) for 6 h as indicated. For the combination, AICAR was added during the last hour of Baf. A1 treatment. Lysates were immunoblotted with the indicated antibodies ($n = 1$). **e** NIH3T3 cells were transfected with siRNA against Cyclin Y or control siRNA and stimulated with 0.5 mM AICAR/50 μM A769662 (A769) or with 200 nM Baf. A1 or a combination. Proteins were analyzed as indicated. ($n = 3$). **f** Representative confocal images of the NIH3T3 cells treated as in panel **e**. Endogenous LC3 was stained with the 4E12 antibody to depict autophagy. Scale bar: 50 μm. **g** LC3 dots as shown in panel **f** were quantified. Statistical significance was measured via unpaired and two-tailed Student's $t$-tests and is presented as follows: **$p < 0.01$, ***$p < 0.001$. All error bars indicate SD. ($n = 3$; 100 cells counted for each replicate; AICAR/A769: t = 9.494, df = 6; Baf. A1: t = 6.549, df = 6; AICAR/A769 + Baf. A1: t = 9.484, df = 4). n biological independent replicate. SD standard deviation. Source data are provided as a Source Data file.

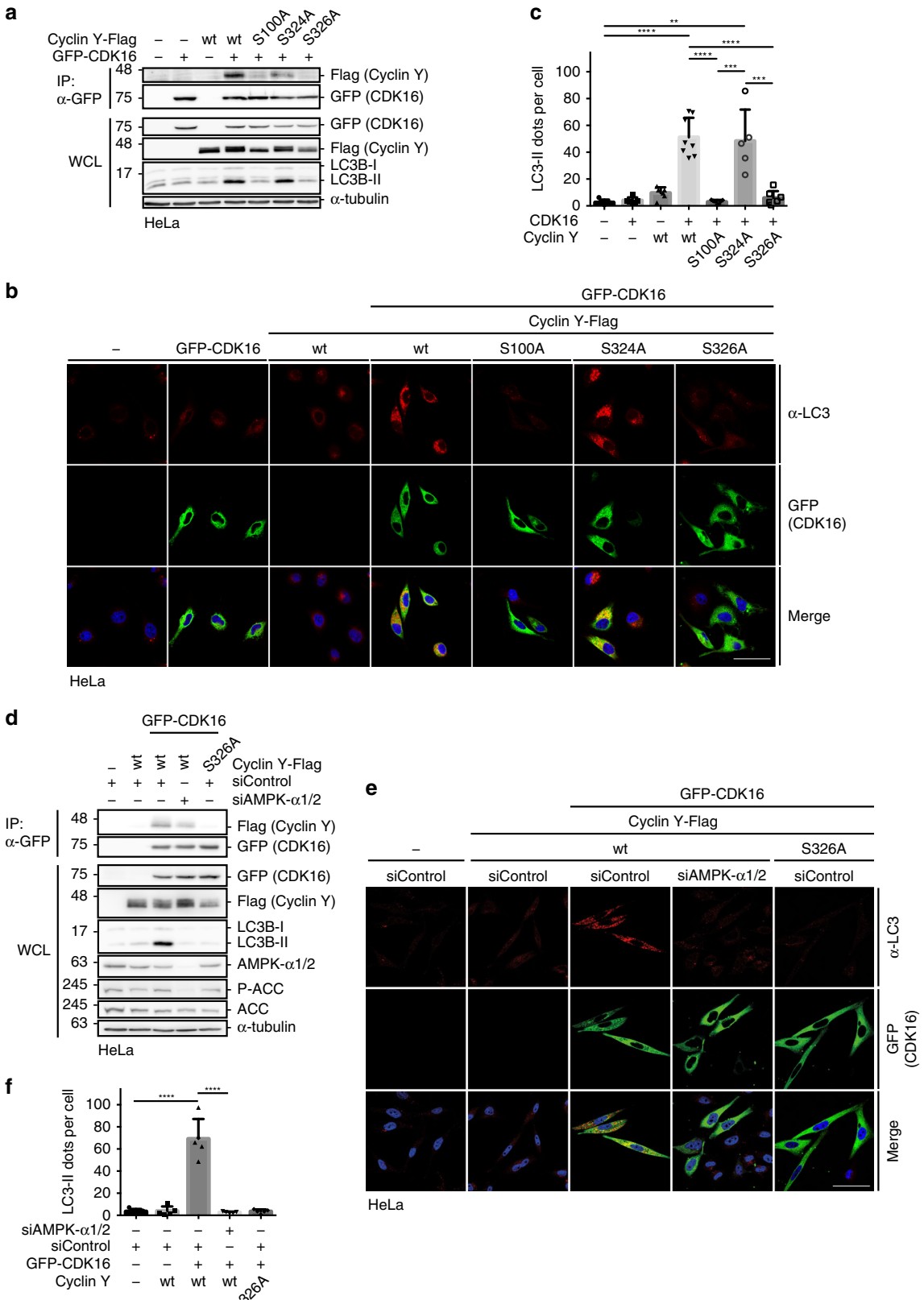

**GST-tagged protein expression and purification.** Purifications of GST-tagged proteins were generally performed using the *E.coli* BL21 strain. After transformation of the construct of interest several colonies was picked to inoculate a starter culture in 50 ml LB medium supplemented with the appropriate antibiotic and 0.5% glucose. This culture was allowed to grow overnight at 37 °C and horizontal shaking at 160 rpm and the next day 500 ml LB medium (with antibiotic and glucose) were inoculated with 25 ml of the starter culture. When the OD$_{600}$ reached the range between 0.5–0.7, protein expression was induced by the addition of IPTG to a final concentration of 0.5 mM and the culture was incubated for another 4 h at 37 °C. Bacteria were harvested by centrifugation (4500 × g, 4 °C, 10 min) and resuspended in 30 ml ice-cold TNE buffer (20 mM Tris base (pH 8.0), 150 mM NaCl, 1 mM EDTA (pH 8.0), 5 mM DTT, 1 mM Pefabloc, 14 µg/ml Aprotinin). After lysis on ice for 30 min with 100 µg/ml lysozyme, proteins were solubilized by sonication and lysates were cleared by centrifugation at 10,000 × g at 4 °C for 30

**Fig. 7 AMPK-dependent phosphorylation of Cyclin Y at S326 is required for autophagy. a** HeLa cells were transfected with GFP-CDK16, Cyclin Y-Flag or phosphosite mutants as indicated. The interaction of CDK16 and Cyclin Y was analyzed in GFP-specific immunoprecipitations (IP). Proteins were analyzed by immunoblotting as specified (WCL). ($n = 2$). **b** Representative confocal images of HeLa cells from panel **a**. GFP-CDK16 identified transfected cells and staining for endogenous LC3 (red, antibody 4E12) monitored autophagy. Scale bar: 50 μm. **c** Quantification of LC3 dots per cell treated as displayed in panel **b**. Statistical significance was measured via unpaired and two-tailed Student's $t$-tests and is presented as follows: $**p < 0.01$; $***p < 0.001$; $****p < 0.0001$. All error bars indicate SD. ($n = 1$; 100 cells were analyzed for each treatment; control vs. Cyclin Y/CDK16: t = 7.274, df = 11; control vs. Cyclin Y S324A/CDK16: t = 4.356, df = 8; Cyclin Y/CDK16 vs. Cyclin Y S100A/CDK16: t = 8.644, df = 13; Cyclin Y/CDK16 vs. Cyclin Y S326A/CDK16: t = 7.765, df = 13; Cyclin Y S100A/CDK16 vs. Cyclin Y S324A/CDK16: t = 5.214, df = 10; Cyclin Y S324A/CDK16 vs. Cyclin Y S326A/CDK16: t = 4.734, df = 10). **d** HeLa cells were transfected with GFP-CDK16 and Cyclin Y-Flag-wt or the S326A mutant and siRNA against AMPK-α1/2 or control. The interaction of CDK16 and Cyclin Y was analyzed in GFP-specific immunoprecipitations. Proteins were analyzed by immunoblotting as specified. ($n = 2$). **e** Representative confocal images of the HeLa cells from panel **d**. GFP-CDK16 identified the transfected cells and staining for endogenous LC3 (red, antibody 4E12) measured autophagy. Scale bar: 50 μm. **f** Quantification of the LC3 dots shown in panel **e**. Statistical significance was measured via unpaired and two-tailed Student's $t$-tests and is presented as follows: $****p < 0.0001$. All error bars indicate SD. ($n = 1$; 250 cells were analyzed for each treatment; siControl vs. siControl Cyclin Y/CDK16: t = 8.113, df = 8; siControl Cyclin Y/CDK16 vs. siAMPK-α1/2 Cyclin Y/CDK16: t = 8.290, df = 8). n biological independent replicate. SD standard deviation. Source data are provided as a Source Data file.

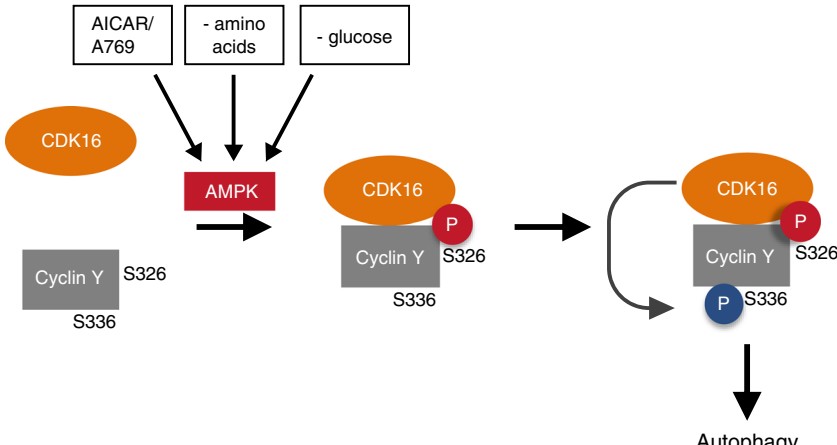

**Fig. 8 Model of the AMPK-dependent activation of Cyclin Y/CDK16 for the induction of autophagy.** AMPK is activated under energy stress situations (-amino acids; - glucose) or by allosteric activators (AICAR/A769) and phosphorylates Cyclin Y at S326. This phosphorylation allows the interaction of Cyclin Y with CDK16 and stimulates the kinase activity of CDK16, as illustrated by the phosphorylation of Cyclin Y at S336. The CDK16 activity downstream of AMPK is required for the efficient induction of autophagy.

min. For purification the lysates were incubated with equilibrated glutathione beads (Glutathione Sepharose 4B, Sigma-Aldrich) for 1 h at 4 °C on a rotation device. Afterwards, beads were pelleted by centrifugation with $500 \times g$ at 4 °C for 2 min, washed three times with 10 ml ice-cold PBS and transferred to a chromatography column (Poly-Prep, BioRad). Columns were washed once more with 1 ml GST wash buffer (100 mM Tris base (pH 8.0), 120 mM NaCl) and GST-proteins were eluted in three fractions using 300 μl GST elution buffer (20 mM glutathione in GST wash buffer) each.

**His₆-tagged protein expression and purification**. His₆-tagged proteins were expressed by a Baculovirus expression system in Sf9 insect cells. The virus production and transfection of the insect cells were carried out with the BaculoGold system (BD Biosciences) according to the manufacturer's guidelines and the following adaptions. $1 \times 10^6$ cells were seeded in 10 ml complete medium in a T75 flask and transfected with TransIT-LT1 transfection reagent (Mirus) after the cells had settled on the bottom. Therefore 200 μl medium were mixed with 10 μl LT1 and 2 μg pLV1392-CCNYiso1-His₆ plasmid were mixed with 0.1 μg BaculoGold DNA. Both mixtures were incubated at room temperature for 10 min, afterwards mixed together and incubated for another 5 min. Before the transfection mixture was added dropwise to the cells, the present medium was changed to FCS-free medium. The cells were incubated for 4 h, then washed with FCS-free medium and finally 5 ml complete medium was added. After 7–10 days the supernatant was filtered through a 0.22 μm PVDF filter (Merck Millipore) and added to freshly seeded cells. To check for protein expression of this first amplification round the remaining cells were used. For the final purification approach $2 \times 10^7$ cells were seeded in 30 ml complete medium in T175 flasks. Each flask was infected with 1 ml filtered, virus-containing supernatant, including approximately $1 \times 10^8$ plaque forming units.

After infection cells were pelleted by centrifugation and resuspended in 10 ml IMAC lysis/wash buffer (20 mM HEPES (pH 7.5), 300 mM NaCl, 10% glycerol,

0.1% NP-40, 5 mM Imidazole, 2 mM β-mercaptoethanol, 1 mM Pefabloc, 1 μg/ml Pepstatin A, 14 μg/ml Aprotinin). The lysate was sonicated and debris were pelleted by centrifugation at $10,000 \times g$ at 4 °C for 30 min. The supernatant was incubated with equilibrated TALON Metal Affinity Resin (Clontech) for 1 h at 4 °C under permanent rotation. Beads were pelleted by centrifugation ($500 \times g$, 4 °C, 2 min), washed thoroughly three times with 5 ml IMAC buffer and finally bound proteins were eluted in 500 μl IMAC elution buffer (20 mM HEPES (pH 7.5), 200 mM NaCl, 10% glycerol, 0.1% NP-40, 10 mM EDTA, 14 μg/ml Aprotinin) by overnight incubation at 4 °C.

**Generation of antibodies**. Phosphosite-specific rabbit or sheep polyclonal antibodies targeting human CDK16 were generated with the help of phospho-peptides by YenZym Antibodies, MRC-PPU Reagents Services team at the University of Dundee or Eurogentec, and affinity purified against the indicated antigens.

YenZym Antibodies generated the rabbit antibodies targeting human CDK16 phospho-S65 (residues 58-SARGPLS-pS-APEIVH-71), phospho-S153 (residues 147-RRLRRV-pS-LSEIGFG-160), and phospho-S155 (residues 149-LRRVSL-pS-EIGFGKL-161). At the University of Dundee the sheep CDK16 phospho-S12 antibody (residues 8-KRQS-pS-MTLRGG-18) and the rabbit phospho-S119 antibody (residues 114-INKRL-pS-LPADI-124) were generated. The rabbit phospho-S461 antibody (residues 453-IHKLPDTT-pS-IFA-464) was generated by Eurogentec.

**Western blotting**. Cells were harvested after stimulation or transfection and then lysed in 400 μl Frackelton buffer for 10 cm dishes (10 mM HEPES (pH 7.5), 50 mM NaCl, 30 mM NaF, 30 mM Na-pyrophosphate, 0.1 mM ZnCl₂, 0.2% Triton X-100, 10% glycerol, protease-inhibitor cocktail (AppliChem)); when phospho-proteins were analyzed additional phosphatase-inhibitors (20 mM beta-glycerophosphate, 1 mM Na₃VO₄, 50 nM ocadaic acid) were added. Lysates were cleared by

**Table 1 Antibodies used for western blots.**

| Antibodies | Used dilution | Vendor |
|---|---|---|
| rabbit polyclonal anti-pan-14-3-3 (K-19) | 1:1000 | Santa Cruz, sc-629; RRID: AB_2273154 |
| rabbit monoclonal anti-ACC (C83B10) | 1:1000 | CST, 3676; RRID: AB_2219397 |
| rabbit monoclonal anti-phospho-ACC (Ser79) (D7D11) | 1:1000 | CST, 11818; RRID: AB_2687505 |
| mouse monoclonal anti-actin (C4) | 1:200 | MP Biomedicals 0869100; RRID: AB_2335304 |
| rabbit monoclonal anti-AMPK-α (D5A2) | 1:1000 | CST, 5831; RRID: AB_10622186 |
| rabbit monoclonal anti-phospho-AMPK-α (Thr172) (40H9) | 1:1000 | CST, 2535; RRID: AB_331250 |
| rabbit monoclonal anti-AMPK-β1/2 (57C12) | 1:1000 | CST 4150; RRID: AB_560860 |
| rabbit polyclonal anti-CDK16 (PCTAIRE-1) (C-16) | 1:1000 | Santa Cruz sc-174; RRID: AB_2158996 |
| rabbit polyclonal anti-CDK16 (PCTAIRE-1) | 1:1000 | Sigma-Aldrich, HPA001366; RRID: AB_1079584 |
| sheep polyclonal anti-phospho-CDK16 (Ser12) | 0,1 µg/ml | This paper (MRC-PPU, University Dundee) |
| rabbit polyclonal anti-phospho-CDK16 (Ser65) | 0,1 µg/ml | This paper (YenZym) |
| rabbit polyclonal anti-phospho-CDK16 (Ser119) | 0,1 µg/ml | This paper (MRC-PPU, University Dundee) |
| rabbit polyclonal anti-phospho-CDK16 (Ser153) | 0,1 µg/ml | This paper (YenZym) |
| rabbit polyclonal anti-phospho-CDK16 (Ser155) | 0,1 µg/ml | This paper (YenZym) |
| rabbit polyclonal anti-phospho-CDK16 (Ser461) | 0,1 µg/ml | This paper (YenZym) |
| rabbit polyclonal anti-Cyclin Y | 1:1000 | Bethyl, A302-376A; RRID: AB_1907259 |
| mouse monoclonal anti-Cyclin Y (2C9E3) | 1:2000 | Proteintech, 66865-1-Ig |
| rabbit polyclonal anti-phospho-Cyclin Y (Ser326) | 0,25 µg/ml | Shehata et al., 2015[15] |
| rabbit polyclonal anti-phospho-Cyclin Y (Ser336) | 0,25 µg/ml | Shehata et al., 2015[15] |
| rabbit polyclonal anti-DYKDDDDK tag | 1:1000 | CST 2368; RRID: AB_2217020 |
| mouse monoclonal anti-FLAG (M2) | 1:5000 | Sigma-Aldrich, F3165; RRID: AB_259529 |
| mouse monoclonal anti-GAPDH (4G5) | 1:500 | Bio-Rad, MCA4740; RRID: AB_2107457 |
| mouse monoclonal anti-GFP (9F9.F9) | 1:2000 | Rockland, 600-301-215; RRID: AB_218216 |
| mouse monoclonal anti-HA | 1:1000 | Covance, MMS-101R; RRID: AB_10064220 |
| rabbit polyclonal anti-LC3B | 1:1000 | CST, 2775; RRID: AB_915950 |
| guinea pig polyclonal anti-p62/SQSTM1 | 1:1000 | Progen, GP62-C; RRID: AB_1542690 |
| mouse monoclonal anti-α-tubulin (B-5-1-2) | 1:5000 | Sigma-Aldrich, T5168; RRID: AB_477579 |
| goat polyclonal anti-guinea pig IgG (H + L) HRP-conjugated | 1:10,000 | Santa Cruz, sc-2438; RRID: AB_650492 |
| AffiniPure rat polyclonal anti-mouse IgG (H + L) HRP-conjugated | 1:10,000 | Jackson ImmunoResearch, 415-035-166; RRID: AB_2340269 |
| AffiniPure goat polyclonal anti-rabbit IgG (H + L) HRP-conjugated | 1:10,000 | Jackson ImmunoResearch, 111-035-144; RRID: AB_2307391 |
| rabbit monoclonal anti-ULK1 (D8H5) | 1:1000 | CST, 8054; RRID: AB_11178668 |
| rabbit monoclonal anti-phospho-ULK1 (S757) (D7O6U) | 1:1000 | CST, 14202; RRID: AB_2665508 |
| rabbit monoclonal anti-Beclin1 (D40C5) | 1:1000 | CST, 3495; RRID: AB_1903911 |
| rabbit monoclonal anti-Raptor (24C12) | 1:1000 | CST, 2250; RRID: AB_561245 |
| rabbit polyclonal anti-phospho-Raptor (S792) (24C12) | 1:1000 | CST, 2083; RRID: AB_2249475 |

centrifugation (15,700 × g, 4 °C, 30 min), protein concentration determined by Lowry Assay (Bio-Rad DC Proteinassay Kit II) and equal amounts of protein were separated by 7.5–15% SDS polyacrylamide gel electrophoresis (SDS-PAGE) followed by transfer onto a nitrocellulose membrane. After incubation with primary and secondary antibodies (listed in Table 1) the proteins of interest were visualized by enhanced chemiluminescence in a LAS-3000 Imager (Fujifilm). Uncropped and unprocessed images of the detected immunoblots are supplied in the Supplementary Source data file and from figshare with the identifier https://doi.org/10.6084/m9.figshare.7077302[68]. Densitometry was performed with ImageJ software.

For the detection of protein phosphorylation by phospho-specific antibodies lysates were loaded twice on two separate gels with equal protein amounts. The nitrocellulose blot of one gel was used to detect the total amount of the protein of interest with a pan antibody, the other blot was used to detect the specific phosphorylation of the protein of interest.

**(Co-) Immunoprecipitation (IP).** For co-immunoprecipitation assays cells were transfected with the indicated constructs and lysates prepared as described above. An aliquot of the lysate was used for expression control (whole cell lysate = WCL). For each immunoprecipitation 10 µl equilibrated Flag-beads (anti-FLAG M2 Affinity Gel, Sigma-Aldrich) were incubated with the rest of the lysates under rotation for 2 h at 4 °C. Afterwards the Flag-beads were washed three times with 1 ml Frackelton buffer and co-precipitated proteins were analyzed by western blotting.

GFP fusion proteins were immunoprecipitated according to the same procedure, using 15 µl of protein G Sepharose slurry (GE Healthcare) with 1 µg anti-GFP antibody (Rockland, 300-301-215) added to the lysates. The immunoprecipitation of endogenous Cyclin Y was carried out with 20 µl of protein A Sepharose slurry (GE Healthcare) and 2 µg anti-Cyclin Y antibody (Bethyl, A302-276A) using the same buffer and protocol. Where indicated, species matched IgG was used as a control IP.

**Immunofluorescence.** Cells were grown on 18 mm round coverslips to about 80% confluence. After the respective treatment cells were fixed with 3.7% paraformaldehyde for 20 min at room temperature and permeabilized with 40 µg/ml Digitonin (Sigma-Aldrich) in PBS for 15 min. Afterwards cells were blocked with 3% goat serum (Sigma-Aldrich) in PBS for 30 min, followed by incubation with the indicated primary antibodies (listed in Table 2) diluted in PBS containing 1% goat serum at 37 °C for 1 h. After three washing steps with 1% goat serum in PBS, cells were incubated with the corresponding secondary antibodies conjugated to Alexa Fluor dyes (listed in Table 2) at 37 °C for 1 h. Samples were then washed with PBS and water and counterstained with Hoechst 33258 (1 µg/ml in H₂O, Sigma-Aldrich), followed by the final mounting with Mowiol 4–88 (Sigma-Aldrich). Immunofluorescent images were obtained using a Zeiss LSM 710 laser-scanning microscope operated by ZEN 2009 LE software.

**Proximity ligation assay (PLA).** U2OS cells were grown on 12 mm round coverslips to about 80% confluence. After the respective treatment cells were fixed with 3.7% paraformaldehyde for 20 min at room temperature and permeabilized with 0.1% Triton-X100 in PBS for 15 min at room temerature. Afterwards cells were blocked with Duolink Blocking solution for 60 min at 37 °C, followed by incubation with the rabbit anti-AMPK-β1/2 antibody (57C12, 1:50) and the mouse anti-Cyclin Y antibody (2C9E3, 1:50) diluted in Duolink Antibody diluant 37 °C for 1 h and two washing steps with TBS plus 0.05% Tween-20. The subsequent incubations with the PLA Plus and Minus probes, ligase and polymerase were performed according to the manufacturer's instructions. The washing steps in between were performed with TBS plus 0.05% Tween-20. Samples were then washed with water and counterstained with Hoechst 33258 (1 µg/ml in H₂O, Sigma-Aldrich), followed by the final mounting with Mowiol 4–88 (Sigma-Aldrich). Immunofluorescent images were obtained using a Zeiss Axioplan 2 microscope with a Plan Neofluar 40×/0.75 objective. Hoechst 33258 staining was detected with the Zeiss filter set 49 and FarRed PLA dots were detected by the Omega Optical filter set XF110-2 with a excitation filter 630/50 and emission filter 695/55. Images were taken with a

**Table 2 Antibodies used for immunofluorescence staining.**

| Antibodies | Used dilution | Vendor |
|---|---|---|
| rabbit monoclonal anti-AMPK-β1/2 (57C12) | 1:50 | CST 4150; RRID: AB_560860 |
| mouse monoclonal anti-Cyclin Y (2C9E3) | 1:50 | Proteintech, 66865-1-Ig |
| mouse monoclonal anti-HA | 1:200 | Covance, MMS-101R; RRID: AB_10064220 |
| mouse monoclonal anti-LC3 (4E12) | 1:50 | MBL, M152-3; RRID: AB_1279144 |
| goat polyclonal anti-mouse IgG (H + L) Alexa Fluor 488 | 1:1000 | Thermo Fisher Scientific, A-11001; RRID: AB_2534069 |
| goat polyclonal anti-mouse IgG (H + L) Alexa Fluor 555 | 1:1000 | Thermo Fisher Scientific, A-21422; RRID: AB_2535844 |
| goat polyclonal anti-mouse IgG (H + L) Alexa Fluor 633 | 1:1000 | Thermo Fisher Scientific, A-21050; RRID: AB_2535718 |
| mouse monoclonal anti-WIPI2 (2A2) | 1:500 | Bio-Rad, MCA5780GA; RRID: AB_10845951 |

CoolSnap HQ[2] camera from Photometrics, operated by the VisiView software from Visitron Systems.

**Settings for immunofluorescence microscopy**. Immunofluorescence images were captured by a Zeiss LSM 710 laser-scanning microscope using a C-Apochromat 40x or 63x water immersion objective. For each picture the bit depth was 8 bit at a resolution of $1024 \times 1024$ pixels and the pinhole was set to 1 airy unit (AU). For single cell pictures a 2.0 digital zoom was applied. The excitation of the EGFP and Alexa Fluor 488 fluorochrome (emission maximum: 507/525 nm) was mediated at a wavelength of 488 nm by an argon laser using a 488 nm single channel PTM and a bandpass filter of 495–550 nm. The fluorochromes mCherry and Alexa Fluor 555 (emission maximum: 610/580 nm) were excited at a wavelength of 561 nm by a helium-neon-laser and detected using a 562–630 nm bandpass filter. The excitation of the Alexa Fluor 633 fluorochrome (emission maximum: 639 nm) was mediated at a wavelength of 633 nm by a helium-neon-laser and detected using a bandpass filter of 635–690 nm. The nuclei staining by Hoechst (emission maximum: 455 nm) was excited by UV-Laser at a wavelength of 352 nm and detected using a 454–553 nm bandpass filter. The detection of endogenes Wipi2b dots in the CDK16$^{+/+}$ and CDK16$^{-/-}$ MEFs was performed with the following settings. Alexa Fuor 488 was excited by an argon laser at 488 nm and fluorescence detected in a range of 493–630 nm. Hoechst 33259 was excited at 405 nm and fluorescence detected in a range of 410–503 nm. PLA dots were automatically counted as the LC3 dots, by ImageJ details are described in the image analysis chapter.

**Kinase assays**. *Kinase assay with recombinant proteins*: To assess AMPK kinase activity in vitro kinase assays were performed in AMPK kinase reaction buffer (10 mM HEPES (pH 7.2), 5 mM MgCl$_2$, 1 mM DTT, 200 μM ATP, 40 μM AMP), including 1 μCi of [γ-$^{32}$P]-ATP (specific activity 3000 Ci/mmol, Hartmann Analytic) per kinase reaction. The reaction mixtures were prepared containing 1–10 ng recombinant AMPK-α1, β1, γ1 and 2 μg of substrate (recombinant GST-/His$_6$-tagged proteins) and incubated at 37 °C for 20 min. The kinase reaction was stopped by addition of 4x sample buffer (320 mM Tris base (pH 6.8) 40% glycerol, 20% SDS, 0.5% bromophenol blue, 200 mM β-mercaptoethanol), the samples were boiled, separated by SDS-PAGE and gels were stained with Coomassie blue. The incorporated radioactive label and thus the phosphorylation was detected by exposure of the previously dried gels to radiosensitive films (FujiFilm SuperRX). Alternatively, samples were analyzed by Western blot using a phospho-specific antibody (anti-phospho-Cyclin Y pS326), in this case instead of using [γ-$^{32}$P]-ATP equal amounts of cold ATP were used in these reaction mixtures.

*Kinase assay with immunoprecipitated CDK16*: HeLa cells were transiently transfected with plasmids encoding GFP-CDK16 and Cyclin Y-Flag and stimulated as indicated. Prior to cell lysis, cells were washed with ice-cold PBS containing 100 μM Na$_3$VO$_4$ and 10 mM NaF. Immunoprecipitation was carried out with 1 μg anti-GFP antibody (Rockland, 300-301-215) and 20 μl of protein G Sepharose slurry (GE Healthcare) at 4 °C for 2 h. Thereafter the beads were washed twice with Frackelton buffer and once with buffer C (50 mM Tris Base (pH 7.5), 15 mM MgCl$_2$) to equilibrate beads to kinase assay conditions. For later visualization of the immunoprecipitation efficiency, half of the beads were kept for direct Western blot analysis. For the kinase reaction beads were incubated in a 30 μl reaction mixture containing CDK16 kinase reaction buffer (100 mM MOPS (pH7.2), 100 mM NaCl, 5 mM MgCl$_2$, 5 mM MnCl$_2$, 1 mM DTT, 20 μM ATP), 1 μCi of [γ$^{32}$P]-ATP (specific activity 3000 Ci/mmol, Hartmann Analytic) and 1 μg MBP (Sigma-Aldrich) as a substrate, at 37 °C for 30 min. After stopping the reaction with sample buffer, the samples were analyzed by SDS-PAGE and the gels stained with Coomassie blue. The incorporated $^{32}$P-label was detected by exposure of the previously dried gels to radiosensitive films (FujiFilm SuperRX).

**Protein microarray**. For the purpose of identifying AMPK substrates, the ProtoArray Human Protein Microarray v5.0 (Invitrogen) was used in an in vitro approach, applying recombinant active AMPK complexes together with radiolabeled ATP to mark modified proteins. The assay was performed according to the manufacturer's guidelines. Briefly, recombinant active His$_6$-α1, β1, γ1 AMPK was prepared by co-expression with LKB1-MO25-STRADα in Rosetta(DE3) *E. coli*

(Merck), lysed in lysis buffer (50 mM NaH$_2$PO$_4$, 30% glycerol, 0.5 M sucrose, 10 mM imidazole, pH 8.0), and protein was purified using nickel-Sepharose HP (GE Healthcare) by washing with the same buffer at 20 mM imidazole and eluting with 250 imidazole concentration[16]. Protein microarrays were first equilibrated at 4 °C for 15 min prior to blocking with 1% BSA in PBS for 1 h at 4 °C under continuous shaking. Immediately prior to use active AMPK was diluted in kinase buffer (20 mM HEPES (pH 7.4), 5 mM MgCl$_2$, 0.1% NP-40, 1% BSA, 1 mM DTT) to a final concentration of 50 nM. The AMPK-containing reaction mixture and the kinase-free control (kinase buffer only) were supplemented with [γ-$^{33}$P]-ATP (3000 Ci/mmol, Hartmann Analytic) to obtain radiolabeled ATP at 33 nM. The assay was started by overlaying the microarray surface with 120 μl of the reaction mixtures. Coverslips ensured even distribution of the solution. Arrays were transferred into a 50 ml reaction tube and incubated at 37 °C for 30 min. After washing twice with 0.5% SDS, two subsequent washing steps were performed with ultrapure water (15 min at room temperature without shaking for each washing step). The arrays were dried by centrifugation for 2 min at $200 \times g$, covered with a plastic wrap and exposed to Hyperfilm MP autoradiography films (GE Healthcare) for 3 h. After development the autoradiographs were scanned using a conventional flatbed scanner (Epson Perfection V700 Photo). The scanned autoradiographs were evaluated using the ProtoArray Prospector v5.2 software (Invitrogen). First the control and AMPK signals were separately quantified by densitometry applying default settings (i.e., background subtraction, signal scatter compensation, outlier detection, and 'calculate Z-factor on per subarray' were all enabled). Then the data were compared again using the ProtoArray Prospector v5.2 software. The results file was opened in Microsoft Excel, signal intensity differences were calculated (i.e., ΔI = 'AMPK signal' - 'control signal') and proteins were sorted according to their ΔI with highest ΔI on top. Additionally, intensity ratios (r) were calculated for each protein (i.e., r = 'AMPK signal'/'control-signal'). A cutoff value for ΔI > 10,000 was arbitrarily chosen in order to restrict the analysis to higher probability AMPK targets. Data quality was further assessed according to the manufacturers suggestions, i.e., Z-Factors greater than 0.4, Z-Scores above 3, and coefficients of variation (CVs) for signals from two replicates less than 0.5.

**Mass spectrometry**. In vitro kinase assays were performed with recombinant His$_6$-Cyclin Y isoform 1 (Q8ND76-1) or GST-CDK16 isoform 1 (Q00536-1) and AMPK as described above. For analysis of CDK16 phosphorylation in cells, HeLa cells were transfected with vectors expressing GFP-CDK16-wt. Cells were treated with 0.5 mM AICAR/50 μM A769662 for 1 h and afterwards lysed as described before and GFP-CDK16 was immunoprecipitated with a GFP antibody. In both cases samples were then boiled in sample buffer and separated by SDS-PAGE. Cyclin Y and CDK16 bands were excised and subjected to digestion with Trypsin as described previously[69]. In brief, the gel pieces containing Cyclin Y and CDK16 were excised, destained, washed with 50 mM Ammoniumbicarbonate (ABC) followed by Acetonitrile (repeated once). After drying of the gel pieces, the proteins were reduced using 10 mM DTT for 45 min at 56 °C and subsequently alkylated using 55 mM Iodoacetamide (RT, in the dark, 30 min). The samples were then washed once more with 50 mM ABC and acetonitrile and dried in a SpeedVac. The proteins were digested overnight with 30 μl (12.5 ng/μl) of mass spectrometry grade Trypsin (Thermo Fisher Scientific). Subsequently, the resulting peptides were extracted from the gel pieces using 50% acetonitrile in 0.1% trifluoroacetic acid and desalted using homemade C18-Tips. Lyophilized peptides were resuspended in 10% formic acid and subjected to nano LC-MS/MS analysis using a nano Ultimate 3000 liquid chromatography system and an Orbitrap Elite or Q Exactive plus mass spectrometer (both Thermo Fisher Scientific). Trapping of the peptides was performed on a precolumn (Acclaim PepMap100, C18, 5 μm, 100 Å, 300 μm i.d. x 5 mm, Thermo Fisher Scientific) for 10 min with buffer A (0.1% formic acid). RSLCnano-Orbitrap Elite: The peptides were then separated on an analytical column (Acclaim PepMap100, C18, 5 μm, 100 Å, 75 μm i.d. × 25 cm) employing a 70 min gradient (0–10 min: 5% buffer B (80% acetonitrile, 0.1% formic acid), 10–45 min: 10–45% buffer B, 45–47 min: 45–99% buffer B; 47–53 min 99% buffer B; 53–70 min 5% buffer B) at 250 nl/min. RSLCnano-Q Exactive plus: the peptides were separated on an Easyspray C18 analytical column (Thermo Scientific) coupled to the Easyspray source (2 μm particle size, 75 μm inner diameter, 50 cm length, 40 °C column oven temperature, 2 kV; Thermo Scientific) using a 90 min gradient:

0–10 min: 5% buffer B (80% ACN/0.1% FA), 10–45 min: 5 to 35% B, 45–55 min: 35 to 50% B, 55–58 min: 50 to 90% B, 58–63 min: 90% B, 63–64 min 99 to 5% B, 64–90 min 5% B.

The mass spectrometers were operated in data-dependent mode. Orbitrap Elite: Full-scan MS spectra were acquired in the Orbitrap ($m/z$ 350–1500) at a resolution of 120,000 and an AGC of 1E6 ions. CID fragmentation in the ion trap was performed on the top 5 precursors of each full scan with collision energy of 35%. Q Exactive plus: resolution of 70,000, AGC of 3E6 ions, scan range 300–1750 m/z. dd-MS$^2$ settings: resolution of 17,500, AGC target 2e5, top 10 precursor fragmentation, collision energy: 27. Raw data were analyzed by using MaxQuant v1.5.1.2 and searched against the human Uniprot database version 06/2015 (canonical and isoforms) using the Andromeda search engine with default mass tolerance settings[70]. Trypsin was set as the protease with two allowed missed cleavages. As the fixed modification Carbamidomethylation (Cys) was set; variable modification settings: Oxidation (Met), Phosphorylation (Ser, Thr, Tyr) and N-terminal protein acetylation. False discovery rates: 0.01 for peptides, proteins and modification sites; minimum peptide score for modified peptides: 40; minimum peptide length: seven amino acids. Further data analysis and reduction for the generation of Supplementary Table 2 was carried out using Perseus[71].

Experiments were performed twice and every sample was measured in duplicate. Intensities were calculated as the mean of all measurements. From these mean values, the ratios of the samples ± AMPK were calculated and normalized against the respective protein intensity ratios. To be taken into consideration for further analysis the following criteria for individual phospho-sites had to be fulfilled: (i) the phosphorylation site had to be identified in at least three out of four runs (either minus or plus); (ii) the minimum phosphosite localization probability ("Localization prob") had to be 0.75 or higher; and (iii) a minimum of average intensity of $1 \times 10^7$ was required. Values for the average intensity were calculated from the "Phospho (STY)Sites.txt" file and "evidence.txt" files as indicated in Supplementary Table 2 from the intensity columns minus AMPK (e.g., "Intensity minus AMPK1") or plus AMPK (e.g., "Intensity plus AMPK1"). AMPK1 and AMPK2 are technical replicates of the first biological experiment. AMPK3 and AMPK4 are technical replicates of the second biological experiment.

The peptide coverage of our mass spectrometry analysis is summarized in the "peptide.txt" files deposited in PRIDE. The protein coverage was determined by comparing the covered peptides to the human Cyclin Y isoform 1 (Q8ND76-1) and human CDK16 isoform 1 sequence (Q00536-1). The in vitro mass spectrometry analysis of Cyclin Y and CDK16 cover 90 and 82% of the protein sequence, respectively. The in vivo mass spectrometry analysis of CDK16 covers 83% of the protein sequence. All analysis cover all potential AMPK sites present in Cyclin Y and CDK16 regarding to the consensus sequences depicted in Supplementary Table 1.

The mass spectrometry data have been deposited to the ProteomeXchange Consortium via the PRIDE partner repository with the dataset identifier PXD009968.

**Quantification and statistical analysis**

*Image analysis.* Counting of the LC3 and PLA dots was performed with the ImageJ software 1.51 using the 'Particle analysis function'. For each condition pictures were taken from random sections of the immunofluorescence coverslips and at least a minimum of 50 cells was analyzed per treatment. Analysis required an 8-bit binary image and a threshold was set to tell the LC3 or PLA dots apart from background staining. A manual threshold was set individually for each experimental setup but identical settings were applied to all pictures of the same experiment to ensure comparability. To analyze the threshold image the 'Analyze particles' algorithm was applied and the minimum size and maximal pixel area size was set from 0.1–1000. For each image the ratio of LC3 or PLA dots/cell were calculated taking the nuclei in the respective image section into account. Values were transferred into GraphPad Prism v6 software and data were presented as bar graphs showing mean values ± SD. Images of the PLA analysis were processed with ImageJ. To avoid monitoring the Hoechst staining in the FarRed channel the minimum setting of the brightness was set to 660 with ImageJ. For a better visualization of the FarRed signal it was converted to green.

*Immunoblot quantification and densitometry.* Band intensity quantification was performed using the ImageJ 1.51 Gel Analyzer Script. For quantification of the autophagy status the LC3-II band density was normalized to actin as loading control, all data points were normalized to the untreated control. Densitometry analysis of the Cyclin Y and CDK16 interaction was performed by calculating the ratio of the co-immunoprecipitated GFP-CDK16 to the total immunoprecipitated Cyclin Y-Flag.

*Statistical analysis.* Statistical parameters and tests are reported in the text and figure legends. If not stated otherwise, all bar graph data show mean values ± SD. Statistical analysis was performed using unpaired and two-tailed Student's $t$-test in the GraphPad Prism v6 software. A p value of $\leq 0.05$ was considered statistically significant. In figures, asterisks denote statistical significance as calculated by Student's $t$-test: *$p < 0.05$; **$p < 0.01$; ***$p < 0.001$, and ****$p < 0.0001$. Please note that the number of experiments (n) given in the figure legends indicates

experiments with identical design. Additional experiments, in part initial testing or measuring additional conditions or set ups, are not indicated, but add up to substantially more results that support the findings we report.

**Reporting summary**. Further information on research design is available in the Nature Research Reporting Summary linked to this article.

## Data availability

The mass spectrometry proteomics data have been deposited at PRIDE with the dataset identifier PXD009968. All other data files that support the finding of this study are available from figshare with the identifier https://doi.org/10.6084/m9.figshare.707302[68]. Source data for Figs. 1–7 and Supplementary Figure 1–7 and Supplementary Table 2 are provided with the paper.

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

## Acknowledgements

We thank B. Lippok and J. Stahl for excellent technical assistance, A. Sechi for technical support and David Stephens, Stefano Ferrari, Terje Johansen, Trond Lamark and Martin Eilers for providing plasmids. The Confocal Microscopy Core Facility of the IZKF Aachen assisted our experiments. This work was supported by the START and IZKF programs of the Medical School of the RWTH Aachen University (J.V.), a VIDI-Innovational Research Grant from the Netherlands Organization of Scientific Research (NWO-ALW Grant no. 864.10.007; D.N), the Chinese Scholarship Council (X.Z.), and a Marie Curie fellowship (Grant PIIF-GA-2012- 332230; D.C.), and a grant from the Deutsche Forschungsgemeinschaft (LU 466/16-2; B.L.).

## Author contributions

Conceptualization, D.N., B.L., and J.V.; Methodology, M.D., D.N., B.L., J.V.; Investigation, M.D., S.K., G.A., C.M.P., X.Z., S.N.S., C.P. J.A., M.B., E.B., and D.C; Validation, M.D. and S.K.; Writing—Original draft, J.V.; Writing—Review and editing, M.D., S.K., C.M.P., J.A. K.S., B.L., D.N., and J.V.; Funding acquisition, B.L., D.N., and J.V.; Resources, E.P., S.G., S.N.S., and K.S.; Supervision, D.N., B.L., and J.V. All authors read and commented on the final version of the manuscript.

## Competing interests

The authors declare no competing interests.
