## [Peer Review File · Nature Communications]

Reviewers' comments:

Reviewer #1 (Remarks to the Author):

Summary

By screening for AMPK substrates, Dohmen et al. identified the Cyclin Y/CDK16 complex as new target. The authors showed that phosphorylation of Serine 326 (S326) of Cyclin Y upon activation of AMPK promotes the binding of Cyclin Y to CDK16 and triggers the activation of the latter. Next, the authors showed that AMPK-activating conditions such as starvation and pharmacological activation of AMPK both induces autophagy in manner dependent on an active Cyclin Y/CDK16 complex. Intriguingly, this signaling axis required ULK1 and BECN1. Together, Dohmen and colleagues provided compelling mechanistic evidence for a novel role of the Cyclin Y/CDK16 complex in regulating autophagy in response to AMPK activation. This work is well controlled and rationalized study. I only have a few concerns.

Major points

- 1) Figure 1C: It would be more meaningful to use other CDKs (e.g. CDK14/15) and cyclins than GST in this in vitro phosphorylation assay.
- 2) In Figure 3H the authors should add the reconstitution of the CDK KO MEFS with the kinase-deficient CDK16-K194R mutant.
- 3) Figure 5 suggests that ULK1 and BECN1 are upstream of Cyclin Y/CDK16 with regard to autophagy induction in response to AMPK activation. Is the phosphorylation of Cyclin Y at S326 and S336 dependent on ULK1 or BECN1? Is Cyclin Y and/or CDK16 a substrate of ULK1?
- 4) The authors should compare Cyclin Y/CDK16-dependent (shown in Figure 6E-G) and ULK1-dependent induction of autophagy in response to AMPK activation. This could be done by adding a panel with ULK1 depletion in Figure 6E. Moreover, it is not clear why the authors do not take advantage of this experimental setting (Figure 6E-G) to reintroduce wild-type as well as S100A, S326A and S336A mutants of Cyclin Y. This would substantially increase the impact of the authors' findings in light of the heavy dependency of this study on overexpression approaches.

Minor points

- 5) The authors should expand the assay from Figure S2A/B to WIPI2 puncta as this is a standard marker for monitoring early autophagic membrane structures.

Reviewer #2 (Remarks to the Author):

The manuscript entitled "AMPK-dependent activation of the Cyclin Y/CDK16 complex controls autophagy" by Dohmen and coworkers describe a new downstream effector AMPK, the CyclinY/CDK16 complex, for inducing autophagy. They describe the phosphorylation of Cyclin Y at S326 as a new substrate for AMPK.

The manuscript is well written and well organized with a large amount of evidence for the biology described.

Minor comments:

The authors write on page 8: "For CDK16 we identified 9 potential in vitro AMPK phosphorylation

sites using mass spectrometry (Table S3). These sites could not be verified in cells because they either did not respond to AMPK activation or were not phosphorylated (Table S3). Thus, none of the in vitro identified AMPK phosphorylation sites of CDK16 appear to be prominent AMPK substrates in cells". What is this statement based on? The +/- A/A ratios? How has these ratios been calculated?

I could not extract any data from the uploaded raw data files in the database search program Proteome Discoverer, so I could not validate the identified phosphopeptides myself. So please provide annotated spectra for identified phosphorylation sites and please provide scores for each phosphopeptide. And please make sure that the uploaded raw data files are readable for the reviewers.

Reviewer #3 (Remarks to the Author):

Dohman et al reported the study of AMPK-CyclinY-CDK16 interaction. The authors showed that Cyclin Y is the substrate of AMPK, and CDK16-cyclinY complex regulate autophagy. I have several comments.

- 1) First of all, the authors did not show the endogenous binding/interaction of AMPK-cyclinY-CDK16. Please add the results of immunoprecipitation and immunofluorescence using NIH3T3 and HeLa cells (Figure 1).
- 2) Fig 2 E and F: Please add the CDK16 and AMPK blots
- 3) Fig 3A: The bands of CyclinY were shifted lower when EBSS was used. Please explain this result. Also, please add the AMPK and p-CyclinY blots. Is the phosphorylation of CyclinY associated with the starvation (use of EBSS)?
- 4) Fig 3E: Please add the blots of CDK16.
- 5) Fig 4: In HeLa cells, the experiments using siRNA/shRNA were missing. The expression levels of CDK16 in HeLa cell is not low. This experiment is important to approach the role of CDK16-related autophagy in malignant cells.
- 6) Please add the graphical abstract (schema).
- 7) DISCUSSION: Reportedly, CDK16 is essential for spermatogenesis and cancer progression. Is the autophagy process of CDK16/CyclinY is associated with spermatogenesis and cancer progression? Please add the discussion on this issue. Also, an important reference describing the study of CDK16/PCTAIRE1 is missing. It has been reported that PCTAIRE1/CDK16 regulates the cancer cell progression via phosphorylation of p27 (Cancer Research, 74(20):5795-807, 2014). The authors should cite this study.

Reviewer #4 (Remarks to the Author):

In this paper from Dohmen et al CyclinY/CDK16 is identified as a downstream target of AMPK important for the induction of autophagy. The authors used a protein microarray of about 9000 proteins to identify CyclinY as a novel AMPK substrate and showed that CyclinY is phosphorylated on Ser326 which leads to activation of the accompanying kinase CDK16. The authors present data suggesting that CyclinY/CDK16 is required for efficient activation of autophagy by AMPK. They further show that overexpression of CyclinY/CDK16 is sufficient to promote autophagy in cells. It is indeed very interesting that CyclinY/CDK16 is reported as a novel inducer of autophagy downstream of AMPK and upstream of ULK1/2. The experiments are for the most part well performed, but there is an almost general lack of quantifications of western blots. Although a novel player has been identified in AMPK-regulated autophagy the study does not provide any insight into substrates of Cdk16-CyclinY that may stimulate autophagy induction.

Hence, the mechanism on how CyclinY/CDK16 can induce increased autophagy is not known.

Specific comments:

1. Fig 1. It would be nice if the authors could verify the AMPK phosphorylation of CyclinY in a cell system where AMPK is activated via the normal route involving LKB1. Will for example inducible activation of LKB1 lead to CyclinY phosphorylation and CDK16 activation?
2. Western blot of P-S236 in FIG2D needs quantification.
3. From the S326 phospho-specific antibody blots there is always a background when AMPK is knocked down and without activation of AMPK. Are there also other kinases that phosphorylate this site? Also, how specific is the phosphospecific antibody for the phosphorylated S326? Usually, there is always a background of phosphorylation of the non-phosphorylated site. Can the authors say anything about this? Is the affinity of the antibody for example more than 100- or 1000-fold higher for the pS326 containing epitope than the non-phosphorylated?
4. Fig 2F really need quantifications of the P-S236 signal(s). What is the correct band here as there are several bands and Cyclin Y seems to be phosphorylated on several sites giving rise to 4-6 bands? The samples should be treated with phosphatase to reveal which of these bands (run on a high resolution SDS PAGE gel) are caused by pS236.
5. Fig 2F: Upon EBSS treatment there is a down-shift perhaps suggesting dephosphorylation upon aa starvation? How do the authors explain this changed migration?
6. Fig3A: Quantifications are needed. There are clear loading differences with more actin in lane 2 for example where also the LC3 II band is strongest.
7. Fig3D quantifications are needed of the blots. The authors should also show p62 levels and they should include the full medium plus Baf. A1 control which is missing.
8. Fig3E quantifications are needed of the blots. The authors should also show p62 levels and they should include the full medium plus Baf. A1 control which is missing.
9. The number of GFP-WIPI-1 puncta per cell in Fig S2A-B is unbelievably high! This cannot be correct. Importantly, a control with cells in full medium is missing. There are antibodies available for staining of endogenous WIPI puncta. I fear overexpression has created an artefact of multiple puncta.
10. Fig 4A and bottom of page 11 it would be more correct say"failed to induce LC3 lipidation (Figure 4A)." Instead of"failed to induce autophagy (Figure 4A)." Since it is actually LC3 lipidation which is shown. Also, the blots should be quantified to measure the generation of LC3 form II (lipidated form).
11. Fig5A BafA1 should be included to measure flux here too. Turnover of p62 could also be measured.
12. What is the role of mTOR in the CyclinY/CDK16-induced autophagy? Will rapamycin increase autophagy in the CDK16 KO cells?

Rebuttal Dohmen et al., “AMPK-dependent activation of the Cyclin Y/CDK16 complex controls autophagy”

First of all, we appreciate the comments by the four reviewers who have carefully read our manuscript and suggested a number of modification / experiments. We have addressed all the comments point-by-point below and think that we have improved the manuscript.

We want to apologize that it took a rather long time to revise the manuscript. The most apparent reasons are that the three first authors are no longer working on the project. Dr. Marc Dohmen has moved to health science administrative position, Ms. Sarah Krieg is presently writing her Doctoral thesis on a project involving the role of ADP-ribosylation in host-virus interaction, and Dr. Agalaridis is a technical application specialist at Miltenyi Biotech. Thus, it was not straightforward to further develop the project. Therefore, we have involved a new PhD student, Jan Amelang, in the project and he has helped in the new experiments that we included in the revised manuscript. Jan has been added to the list of authors on our paper.

Reviewers' comments:

Reviewer #1 (Remarks to the Author):

Summary

By screening for AMPK substrates, Dohmen et al. identified the Cyclin Y/CDK16 complex as new target. The authors showed that phosphorylation of Serine 326 (S326) of Cyclin Y upon activation of AMPK promotes the binding of Cyclin Y to CDK16 and triggers the activation of the latter. Next, the authors showed that AMPK-activating conditions such as starvation and pharmacological activation of AMPK both induces autophagy in manner dependent on an active Cyclin Y/CDK16 complex. Intriguingly, this signaling axis required ULK1 and BECN1. Together, Dohmen and colleagues provided compelling mechanistic evidence for a novel role of the Cyclin Y/CDK16 complex in regulating autophagy in response to AMPK activation. This work is well controlled and rationalized study. I only have a few concerns.

We thank the reviewer for acknowledging the compelling evidence that we present to define a novel role for the Cyclin Y/CDK16 complex in the regulation of autophagy in response to AMPK activation.

Major points

1) Figure 1C: It would be more meaningful to use other CDKs (e.g. CDK14/15) and cyclins than GST in this in vitro phosphorylation assay.

In Figure 1C we confirm our protein array data, which suggested that AMPK phosphorylates both Cyclin Y and CDK16. Indeed, this is reproduced with GST as control because CDK16 was expressed as GST fusion protein. GST alone does not give a signal indicating that it is not a substrate of AMPK. Therefore, we believe it is fair to say that CDK16 is phosphorylated (and not GST) by AMPK.

We note that several CDKs (CDK4-10 and CDK14) as well as several Cyclins (D2, D3, E2, F,

G2, J and L1) were spotted on the microarray. None of these proteins gave a signal above background. Thus, we think that we have sufficient Cyclin and CDK controls to suggest that the phosphorylation of Cyclin Y and CDK16 is selective. In other words, of the more than 9,000 spotted proteins on the microarray only 63 were significantly labelled by AMPK, indicating a high degree of overall selectivity. Furthermore, the putative functional involvement of CDK14/15 as autophagy regulator has been ruled out as shown in Figure 4D-F.

2) In Figure 3H the authors should add the reconstitution of the CDK KO MEFS with the kinase-deficient CDK16-K194R mutant.

The added benefit of the suggested experiment is not fully clear. Perhaps the reviewer alludes to the possibility that CDK16 activity might be dispensable for driving autophagy, i.e. inactive CDK16 might be of structural relevance (e.g. as a scaffold for Cyclin Y or for other unknown proteins). Our experiments shown in Figs. 4A-C and S3A-C in murine NIH-3T3 and human HeLa cells, respectively, document that CDK16-K194R has no significant activity above control. This is seen both by analyzing the ratio between LC3-II / LC3-I and by staining for LC3 puncta. Thus, we have addressed the role of the CDK16 kinase activity in two different cellular systems with comparable results and argue that we have sufficient evidence to suggest strongly that CDK16 catalytic activity is required to induce autophagy. Please also note that the CDK16-K194R mutant in complex with Cyclin Y is indeed catalytically inactive in kinase assays as shown by us previously (Mikolcevic, ..., Geley, MCB 32, 868, 2012; Shehata, ..., Sakamoto, Cell Signal 24, 2085, 2012). In conclusion, our findings reveal that the catalytic activity of CDK16 is required for the induction of autophagy.

3) Figure 5 suggests that ULK1 and BECN1 are **upstream** of Cyclin Y/CDK16 with regard to autophagy induction in response to AMPK activation. Is the phosphorylation of Cyclin Y at S326 and S336 dependent on ULK1 or BECN1? Is Cyclin Y and/or CDK16 a substrate of ULK1?

*The reviewer interprets our findings such that “ULK1 and BECN1 are **upstream** of Cyclin Y/CDK16 with regard to autophagy induction”. We suggest that Cyclin Y/CDK16 is upstream of ULK1 and Beclin1. Obviously, either of the two interpretations could be correct. Because ULK1 and Beclin1 are necessary for LC3 lipidation, the Cyclin Y/CDK16 complex stimulates lipidation, and the knockdown of ULK1 or Beclin1 decreases LC3 lipidation in the presence of functional Cyclin Y/CDK16, the order of events is not clear at present. Therefore, the data shown in Figure 5 could also be in agreement with Beclin1 and ULK1 acting **downstream** of the Cyclin Y/CDK16 complex. Notably, in Figure 6, it is shown that lack of either Cyclin Y or CDK16 in conjunction with AMPK activation still blocks LC3 lipidation. These findings establish a mandatory role of Cyclin Y /CDK16 in LC3 lipidation (similar to ULK1 and Beclin 1). We acknowledge that further studies are needed to elucidate the exact signaling pathways and interdependency of the various kinases in the regulation of autophagy. For this reason, we stated only that “Cyclin Y/CDK16 appears to require both ULK1 and Beclin1 complexes” in the last sentence of the paragraph on page 13 that refers to figure 5.*

[REDACTED]

4) The authors should compare Cyclin Y/CDK16-dependent (shown in Figure 6E-G) and ULK1-dependent induction of autophagy in response to AMPK activation. This could be done by adding a panel with ULK1 depletion in Figure 6E. Moreover, it is not clear why the authors do not take advantage of this experimental setting (Figure 6E-G) to reintroduce wild-type as well as S100A, S326A and S336A mutants of Cyclin Y. This would substantially increase the impact of the authors' findings in light of the heavy dependency of this study on overexpression approaches.

To study different mutants, in particular of Cyclin Y, which we define as novel AMPK target, but also of CDK16, we rely on over-expression of these proteins. These experiments define the requirements for Cyclin Y/CDK16 to induce autophagy. However, we would like to point out that all our findings are backed by knock-down and knock-out studies, which demonstrate the importance of Cyclin Y/CDK16 for AMPK-dependent induction of autophagy in different cell types.

We have not performed knockdown of ULK1 in cells, in which autophagy was stimulated in response to AICAR/A769. However, we have shown that ULK1 deficiency blocks the LC3 lipidation that is triggered by the active Cyclin Y/CDK16 complex in Figure 5. The essential role of ULK1 downstream of AMPK is very well documented, both using knockdown and knockout studies, for the induction of autophagy. This seems an experiment that is not really adding novel information but being merely confirmatory.

We have used the S100A and the S326A Cyclin Y mutants in several experiments, which collectively demonstrate their functional relevance. Phosphorylation of S100 is important for CDK16 binding and the S100A does not interact with CDK16 (as shown previously by us in Shehata, ..., Sakamoto, Biochemical J. 469, 409, 2015) and as we confirm in Fig. 7. Furthermore, we demonstrate that this mutant does not induce autophagy together with CDK16. This is consistent with the CyclinY-L222A/S224A mutant (CyclinY-AA), which was described by us as CDK16 binding deficient (Fig. S3C and Mikolcevic, ..., Geley, MCB 32, 868, 2012). Similarly to the Cyclin Y-S100A mutant, the CyclinY-AA mutant does not stimulate autophagy (Figs. 4 and S4). Thus, quite clearly our studies with the S100A and the S326A Cyclin Y mutants are consistent with previous work, with the Cyclin Y-AA mutant and with catalytic inactive CDK16 mutant, all demonstrating that the complex and the activity of Cyclin Y and CDK16 is necessary for autophagy induction. Thus, the experiment suggested by the reviewer would be merely confirmatory to an already well established, consistent set of data. But of course we realize that more information about the precise role of different Cyclin Y mutants would be a valuable add on to our findings. Rather than performing more knockdown studies, we opted for knockouts of Cyclin Y and CDK16 in tumor cells with subsequent expression of mutants in a clean background. These preliminary efforts were not very successful as the tumor cells tested seem to be sensitive to the loss of these proteins. This maybe the consequence of autophagy being important in many tumor cells for proliferation. Clearly this needs more effort to establish systems, in which we can address the role of mutants more efficiently. It might require non-transformed cells, which we have not attempted yet but will do in the future.

We have not analyzed the role of the phosphorylation at S336 of Cyclin Y in any detail. We have used it exclusively as a read-out for Cyclin Y/CDK16 kinase activity. Of note is that we have shown previously that the Cyclin Y-S336A mutant interacts with CDK16 and shows efficient kinase activity when isolated from cells (Shehata, ..., Sakamoto, Biochemical J. 469, 409, 2015). Thus, at present the functional relevance of this automodification site is not known. But we agree with the reviewer that a detailed analysis of this phosphorylation site is warranted. Whether it is necessary for the induction of autophagy is not known at present. We feel that analyzing this phosphorylation site is beyond the scope of the manuscript.

Finally, we tested whether ULK1 is capable of phosphorylating Cyclin Y/CDK16. In these preliminary experiments we did neither observe phosphorylation of Cyclin Y nor CDK16, supporting our view that ULK1 is probably downstream of Cyclin Y/CDK16. As these are only in vitro experiments at present, we have not included these findings in the revised manuscript.

Minor points

5) The authors should expand the assay from Figure S2A/B to WIPI2 puncta as this is a standard marker for monitoring early autophagic membrane structures.

We agree that WIPI2 is a good marker to monitor early autophagic membrane structures. With WIPI2, LC3 lipidation, LC3 puncta formation, and p62 turnover we have used four standard markers for measuring autophagy. We note that in our hands the analysis of p62 was not always straightforward, probably because of the enhanced de novo synthesis in response to autophagy induction, as reported by others. Nevertheless, Figure S3A/B shows impaired formation of WIPI2 dots in CDK16 knockout cells as supplementary information to the standard LC3 puncta formation used in many of our experiments. Thus, this experiment supports our general conclusion regarding the role of CDK16 for the induction of autophagy in response to starvation. It is unclear how exactly the reviewer proposes to expand on this assay. In our view it supports our main conclusion, this is what we hoped to achieve and in fact did.

Reviewer #2 (Remarks to the Author):

The manuscript entitled "AMPK-dependent activation of the Cyclin Y/CDK16 complex controls autophagy" by Dohmen and coworkers describe a new downstream effector AMPK, the CyclinY/CDK16 complex, for inducing autophagy. They describe the phosphorylation of Cyclin Y at S326 as a new substrate for AMPK.

The manuscript is well written and well organized with a large amount of evidence for the biology described.

We thank the reviewer for his/her positive evaluation of our manuscript and pointing out that we present “a large amount of evidence for the biology described”.

Minor comments:

The authors write on page 8: “For CDK16 we identified 9 potential in vitro AMPK

phosphorylation sites using mass spectrometry (Table S3). These sites could not be verified in cells because they either did not respond to AMPK activation or were not phosphorylated (Table S3). Thus, none of the in vitro identified AMPK phosphorylation sites of CDK16 appear to be prominent AMPK substrates in cells". What is this statement based on? The +/- A/A ratios? How has these ratios been calculated?

*The in vitro mapping of AMPK phosphorylation sites on CDK16 revealed 8 sites (Table S3, identified **Residues** and **Phosphopeptides**). We then compared the phosphorylation of CDK16 expressed in cells either untreated or stimulated with AICAR/A769662 for 1 hour. We were able to detect 5 of these 8 sites (plus one additional site), for 3 sites we did not obtain signals (indicated with n.d. in Table S3). We then compared the signals \pm AICAR/A769662 and determined the ratio as described in the figure legend of table S3 and as indicated in the newly provided supplementary excel file (Table S4). Phosphorylation at S65 increased two-fold, while all the other sites showed very little variation (Table S3, second to last column). In parallel we analyzed phosphorylation of S326 of Cyclin Y, for which we had an antibody and observed clear effects (e.g. Fig. 2). We also generated phospho-specific antibodies for most of the potential CDK16 phosphorylation sites to complemented the mass spectrometry analysis. We tested the phosphorylation of CDK16 by AMPK in vitro and in cells. The results with the phospho-specific antibodies support the mass spectrometry analysis. CDK16 is an in vitro AMPK substrate but not in cells (Figure S1). We have now modified the text to more clearly describe how the analysis was done and what the base was for concentrating on Cyclin Y-S326.*

I could not extract any data from the uploaded raw data files in the database search program Proteome Discoverer, so I could not validate the identified phosphopeptides myself. So please provide annotated spectra for identified phosphorylation sites and please provide scores for each phosphopeptide. And please make sure that the uploaded raw data files are readable for the reviewers.

We apologize for this problem. We have re-downloaded the data submitted to pride (<http://www.ebi.ac.uk/pride>; PXD009968) using the username: reviewer54398@ebi.ac.uk and password: ZDU0CNer as stated in the manuscript (header Data availability). We have tested all raw data using MaxQuant 1.5.1.2 and 1.6.3.3 as well as the demo version of Proteome Discoverer 2.2. We found that every single file was readable (and indeed able to be analyzed) by all programs and versions used. Therefore, we unfortunately cannot explain the problem experienced by the reviewer. It may be that an older PD version (such as 1.2 or similar) might not reliably recognize and read Q Exactive plus and maybe even Elite files, but we a) do not use PD and b) also do not have access to older PD versions through former colleagues or similar.

We have re-performed the entire analysis and uploaded all MaxQuant output files to PRIDE according to the guidelines from <https://www.ebi.ac.uk/pride/help/archive/faq#MaxQuant> as well as the guidelines from Molecular and Cellular Proteomics regarding annotated spectra <https://www.mcponline.org/page/content/annotated-spectra>. The data has also been checked with and can be visualized by MS-Viewer (<http://msviewer.ucsf.edu/prospector/cgi-bin/msform.cgi?form=msviewer>).

We have now provided scores for all phosphorylation sites in Table S3 and expanded the figure legend and provided an annotated spectrum for CCNY S326 (performed using

PDV, <https://github.com/wenbostar/PDV>; <https://www.ncbi.nlm.nih.gov/pubmed/30169737>) in supplementary Figure S2A. In addition, we have convoluted a excel file (see above, Table S4) containing the relevant information for the identification and relative quantification of the phosphorylation sites. It contains the Perseus output of the Phospho (STY)Sites.txt of the individual experiments file after applying the modification "expand site table"; the unmodified, raw Phospho (STY)Sites.txt of the individual experiments as well as the respective entry of the proteinGroups.txt file of the individual experiments, where the ratio was calculated that was subsequently used for normalization.

Reviewer #3 (Remarks to the Author):

Dohman et al reported the study of AMPK-CyclinY-CDK16 interaction. The authors showed that Cyclin Y is the substrate of AMPK, and CDK16-cylinY complex regulate autophagy. I have several comments.

1)First of all, the authors did not show the endogenous binding/interaction of AMPK-cylinY-CDK16. Please add the results of immunoprecipitation and immunofluorescence using NIH3T3 and HeLa cells (Figure 1).

Indeed, this is true. Early on in this project we have tested whether we can observe interaction of AMPK with Cyclin Y/CDK16. However, under co-IP buffer conditions we were unable to detect interaction. These are conditions that we have used to see a number of interaction (e.g. recently in Bütepage et al., Scientific Reports 2018), documenting that we are able to perform such experiments. However, in our experience, the interactions of enzyme-substrate complexes are more difficult to see and in many instances such interactions are too transient. Although some kinases seem to stably interact with its cognate substrate, the alternative so-called 'hit and run' mode describes kinases that do not stably interact with their substrates. Co-localization using immunofluorescence may be informative but frequently is only demonstrating a certain level of proximity. Thus, in our view such experiments will not add to the overall information that we provide, which includes in vitro phosphorylation and strong in cell correlations of AMPK catalytic activity with Cyclin Y-S326 phosphorylation.

2)Fig 2 E and F: Please add the CDK16 and AMPK blots

We have now added the Western blot for CDK16 in Fig. 2E and for AMPK in Fig. 2F.

3)Fig 3A: The bands of CylinY were shifted lower when EBSS was used. Please explain this result. Also, please add the AMPK and p-CyclinY blots.

We have now added the Western blot for AMPK and for Cyclin Y-S326P in Fig. 3A.

Regarding the mobility shift of Cyclin Y upon treatment with EBSS, we have noticed this in all our experiments. This is independent of whether CDK16 is co-expressed or not. These shifted proteins seem not to be some artificial signals appearing in response to EBSS because these protein bands disappear upon knockdown of Cyclin Y. One possibility that we addressed is an

altered subcellular localization because Cyclin Y is known to undergo lipidation. However, we did not see any altered localization (see Figure 1 below for reviewers, also no difference when co-expressed with CDK16 (not shown)). It is possible that other post-translational mechanisms target Cyclin Y upon treatment with EBSS and thus induce an enhanced mobility of the protein, however, this remains to be studied further.

Figure 1: Immunofluorescence staining of HeLa cells transfected with a Cyclin Y-Flag expression plasmid. Cells were kept under normal growth condition (10% FCS) or were treated for 2 h with EBSS. Cyclin Y was stained with the Flag M2 antibody and an anti-mouse secondary antibody coupled to the Alexa Fluor 555 dye.

Is the phosphorylation of Cyclin Y associated with the starvation (use of EBSS)?

Our findings displayed in Fig. 2F and 3A demonstrate that in addition to EBSS, the activation of AMPK by AICAR/A769662, the removal of glucose, and Ca^{2+} signaling also induce Cyclin Y phosphorylation at S326. Thus, there is a strong link to AMPK but this is not strictly limited to starvation.

4) Fig 3E: Please add the blots of CDK16.

A Western blot showing the lack of signal for CDK16 is shown in Fig. 3E.

5) Fig 4: In HeLa cells, the experiments using siRNA/shRNA were missing. The expression levels of CDK16 in HeLa cell is not low. This experiment is important to approach the role of CDK16-related autophagy in malignant cells.

As pointed out above in the response to reviewer #1, we opted for knockout experiments of Cyclin Y and CDK16 in tumor cells. This would have given us the possibility to introduce mutants of

Cyclin Y and CDK16 for further analysis in a clean background. Unfortunately, these preliminary efforts were not very successful as the tumor cells tested seem to be sensitive to the loss of these proteins. We are now planning to use inducible systems so that we are able to study this in more detail. Clearly, the role of autophagy in many tumor cells will be important to be addressed in more detail. Also in light of several recent publications that have shown that knockdown of CDK16 in tumor cells inhibits proliferation as we pointed out in the discussion (see references 56 – 61 in the revised manuscript). Thus, to study the wider application of Cyclin Y/CDK16 for tumors is important, possibly providing opportunities for interference with selective inhibitors. These experiments are not within the scope of this manuscript.

Moreover, in response to this reviewer’s comment we wanted to use an inhibitor of CDK16 to further document the pro-autophagic effect of the Cyclin Y/CDK16 complex. Unfortunately, these studies are not sufficiently developed for being used here. But to document the effort, we have screened a small molecular weight compound library (some 3000 compounds) and have now identified a few that seem to have some activity (see Figure 2 below for reviewers only). We are in the progress to evaluate selectivity and to further study the positive hits.

Figure 2: NIH3T3 cells were treated for 1 h with potential CDK16 inhibitors. Afterwards the cells were incubated for 2 h in EBSS in the presence of the inhibitors. Induction of autophagy was measured by determine of p62 protein levels and LC3 lipidation.

6) Please add the graphical abstract (schema).

A graphical abstract is now included.

7) DISCUSSION: Reportedly, CDK16 is essential for spermatogenesis and cancer progression. Is the autophagy process of CDK16/CyclinY associated with spermatogenesis and cancer progression? **Please add the discussion on this issue.** Also, an important reference describing the study of CDK16/PCTAIRE1 is missing. It has been reported that PCTAIRE1/CDK16 regulates the cancer cell progression via phosphorylation of p27 (Cancer Research, 74(20):5795-807, 2014). The authors should cite this study.

We thank the reviewer for pointing this out and we have now expanded on the discussion. Of note is that we have also seen phosphorylation of p27 by Cyclin Y/CDK16 and thus have confirmed at least some of the basic information in the indicated Cancer Research paper.

Reviewer #4 (Remarks to the Author):

In this paper from Dohmen et al Cyclin Y/CDK16 is identified as a downstream target of AMPK important for the induction of autophagy. The authors used a protein microarray of about 9000 proteins to identify CyclinY as a novel AMPK substrate and showed that CyclinY is phosphorylated on Ser326 which leads to activation of the accompanying kinase CDK16. The authors present data suggesting that CyclinY/CDK16 is required for efficient activation of autophagy by AMPK. They further show that overexpression of CyclinY/CDK16 is sufficient to promote autophagy in cells.

It is indeed very interesting that CyclinY/CDK16 is reported as a novel inducer of autophagy downstream of AMPK and upstream of ULK1/2. The experiments are for the most part well performed, but there is **an almost general lack of quantifications of western blots.**

We thank the reviewer for acknowledging that defining Cyclin Y/CDK16 as a regulator of autophagy is of interest. We have now quantified our blots and have indicated this information in the revised versions of the figures.

Although a novel player has been identified in AMPK-regulated autophagy the study **does not provide any insight into substrates of Cdk16-CyclinY that may stimulate autophagy induction.** Hence, the mechanism on how Cyclin Y/CDk16 can induce increased autophagy is not known.

Indeed, we have not defined critical substrates for Cyclin Y/CDK16. However, we like to indicate that we started out by asking for effectors of AMPK. This allowed us to define Cyclin Y/CDK16 as a novel AMPK substrate and as a regulator of autophagy. It will certainly be of importance to define the targets of Cyclin Y/CDK16 that disseminate the upstream information. In fact, we have begun to address this issue using protein arrays (as done for AMPK shown in this manuscript) and by using selective proteins that we think are downstream.

[REDACTED]

However, sorting out which of the several sites are functionally relevant is beyond the scope of this manuscript. But clearly, this will be important to analyze and to define the order of events.

On a more general note, the biological functions of CDK16 are still rather ill-defined. We are certainly keen in defining the role of this kinase further and the studies that we have initiated will hopefully provide clues to the functional relevance of this kinase. Identifying Cyclin Y/CDK16 substrates will certainly be an important part of these future analyses.

Specific comments:

1. Fig 1. It would be nice if the authors could verify the AMPK phosphorylation of CyclinY in a cell system where AMPK is activated via the normal route involving LKB1. Will for example inducible activation of LKB1 lead to CyclinY phosphorylation and CDK16 activation?

We realize that HeLa cells do not express LKB1. However, many recent publications have shown that AICAR/A679662 through AMPK is sufficient to induce all the changes associated with autophagy, comparable to cells that express LKB1. AICAR/A679662 does not require any upstream kinase. Moreover, we have used U2OS cells (Fig. S2) and NIH3T3 cells (Fig. 2E and F), in which we observe Cyclin Y-S326P in response to activation of AMPK. These two cell lines are LKB1 positive. The findings are comparable between HeLa and U2OS / NIH3T3. Furthermore, glucose starvation is known to trigger LKB1-dependent AMPK activation, i.e., linking AMPK activity status to energy charge (AMP/ATP ratio) and indeed glucose starvation induces Cyclin Y-S326P in NIH3T3 cells. Also ionomycin induces Cyclin Y-S326P. Ionomycin activates CaMKK2, a kinase which has been described to affect AMPK. Thus, we think that we have ample evidence to suggest that AMPK can be activated in a more classical way, i.e. by integrating LKB1-dependent phosphorylation, but also by other means. We have now added more information regarding this, e.g. in relation to Fig. 2F in the text.

2. Western blot of P-S236 in FIG2D needs quantification.

We have added quantification of this experiment.

3. From the S326 phospho-specific antibody blots there is always a background when AMPK is knocked down and without activation of AMPK. Are there also other kinases that phosphorylate this site?

This is an important question. We have not tested whether other kinases can phosphorylate Cyclin Y-S326. We have now interrogated different databases that allow prediction of potential kinases using the human sequence DLRRSARKRSAS₃₂₆ADNTLPRW (Fig. S2) that might phosphorylate this site. The results were variable depending on the database used, nevertheless besides AMPK consistently AKT was suggested as potential kinase. In addition, different kinases of the AGC kinase group (cAMP-dependent protein kinase (PKA), the cGMP-dependent protein kinase (PKG) and the protein kinase C (PKC) families) were indicated, for example DMPK. Thus, it is well possible that other kinases can also phosphorylate Cyclin Y-S326. It will certainly be interesting to understand in future studies which other kinases can indeed phosphorylate this site in cells and what the functional consequences are of this modification. In particular, is this phosphorylation always associated with autophagy or does it also have functions beyond this process. We are planning to address this in follow-up studies.

3. Also, how specific is the phosphospecific antibody for the phosphorylated S326? Usually, there is always a background of phosphorylation of the non-phosphorylated site. Can the authors say anything about this? Is the affinity of the antibody for example more than 100- or 1000-fold higher for the pS326 containing epitope than the non-phosphorylated?

We agree with the reviewer that this is an important question. We estimated the specificity of the antibody for the phosphorylation at S326 with the help of the in vitro kinase assay in Fig. 2A. We added a longer exposé of the phospho-S326 blot to detect the background signal for Cyclin Y wt in the absence of AMPK or for the Cyclin Y-S326A mutant. The quantification of the phospho-S326 blot revealed an approximate 40-fold signal increase for Cyclin Y wt in the presence of

AMPK compared to all other samples in this experiment. This is only a rough estimation of the specificity of the antibody because we don't know the stoichiometry of the Cyclin Y phosphorylation by AMPK.

4. Fig 2F really need quantifications of the P-S236 signal(s). What is the correct band here as there are several bands and Cyclin Y seems to be phosphorylated on several sites giving rise to 4-6 bands? The samples should be treated with **phosphatase** to reveal which of these bands (run on a high resolution SDS PAGE gel) are caused by pS236.

As indicated in the response to reviewer #3, we do not have a good explanation for the downwards shift of Cyclin Y upon EBSS treatment. We excluded a direct role of CDK16, it seems not to relate to a possible lipidation, and typically phosphorylation reduces mobility (at least in our hands we did not observe such an increased mobility due to phosphorylation). Because these shifted proteins are disappearing in the Cyclin Y knockdown, we assume that they are indeed Cyclin Y. It is possible that other post-translational mechanisms target Cyclin Y upon treatment with EBSS and thus induce an enhanced mobility of the protein, however, this remains to be studied further.

We have now quantified the S326P signals.

5. Fig 2F: Upon EBSS treatment there is a down-shift perhaps suggesting dephosphorylation upon aa starvation? How do the authors explain this changed migration?

Please see discussion to point #4.

6. Fig3A: Quantifications are needed. There are clear loading differences with more actin in lane 2 for example where also the LC3 II band is strongest.

We have now quantified the LC3-II signals.

7. Fig3D quantifications are needed of the blots. The authors should also show p62 levels and they should include the full medium plus Baf. A1 control which is missing.

We have now quantified the LC3 signals. Baf. A1 on the full medium has been done in Fig. 6E in combination with AICAR/A769.

8. Fig3E quantifications are needed of the blots. The authors should also show p62 levels and they should include the full medium plus Baf. A1 control which is missing.

We have now quantified the LC3 signals and added the p62 blot.

9. The number of GFP-WIPI-1 puncta per cell in Fig S2A-B is unbelievably high! This cannot

be correct. Importantly, a control with cells in full medium is missing. There are antibodies available for staining of endogenous WIPI puncta. I fear overexpression has created an artefact of multiple puncta.

Indeed, the WIPI signal is very high. This is because we did not select cell clones for low expression. Rather we analyzed populations of infected cells. Nevertheless, these experiments demonstrate that WIPI puncta are reduced in CDK16 KO cells in response to EBSS treatment, supporting the findings with LC3 lipidation and puncta formation.

We have now tested antibodies against WIPI1 and WIPI2 (specifics are below). All gave significant background in our hands and, as far as we could evaluate the stainings, showed a similar trend. However, the data were not of sufficient quality so that we decided not to use these stainings.

Specifics: Anti-WIPI1 rabbit monoclonal antibody (EPR8110) raised against aa 50 – 150 of human WIPI1 was obtained from Abcam (Catalogue-No. ab128901). Anti-WIPI2 mouse monoclonal antibody (2A2) raised against the C-terminal peptide CSALRLDEDSEHPPMILRTD was obtained by BioRad (Catalogue-No. MCA5780GA).

10. Fig 4A and bottom of page 11 it would be more correct say ...”failed to induce LC3 lipidation (Figure 4A).” Instead of ...”failed to induce autophagy (Figure 4A).” Since it is actually LC3 lipidation which is shown. Also, the blots should be quantified to measure the generation of LC3 form II (lipidated form).

We have modified the text accordingly and quantified the blots.

11. Fig5A BafA1 should be included to measure flux here too. Turnover of p62 could also be measured.

It is not quite clear to us why a flux experiment would add new, important information. This experiment was designed to address whether Cyclin Y/CDK16 requires ULK1 and/or Beclin1 for inducing autophagy. The data from Fig. 5 seem really straightforward. Please note that we have performed flux experiments in combination with Cyclin Y/CDK16-induced autophagy in Fig. 4 A to C and S3.

12. What is the role of mTOR in the CyclinY/CDK16-induced autophagy? Will rapamycin increase autophagy in the CDK16 KO cells?

We have performed preliminary experiments with rapamycin and observed increased autophagy (not shown), as noted by others. However, we have not addressed the role of mTOR in cooperation with Cyclin Y/CDK16. This is certainly an interesting question but we feel that to fully exploit this hypothesis is not in the scope of this manuscript.

Reviewers' comments:

Reviewer #1 (Remarks to the Author):

The authors provided sufficient explanations to address the majority of my concerns. However, a few open issues remain.

Add 2) Figure 4A-C and S3C indeed show that the CDK16's kinase activity is required but these settings are always overexpression approaches. Reconstituting CDK16 KO MEFs with CDK16-K194R would allow to strengthen the authors finding at near endogenous levels and more physiological conditions. Since the authors already used this reconstitution strategy for wild-type CDK16, it should be quite straightforward to expand this to the mutant.

Add 5) In Figure S3A/B the authors monitored WIPI1. However, I asked to expand this assay to WIPI2 as this is one of the gold standards for autophagosome precursor structures.

Besides, LC3-II refers to an immunoblot readout. Please change "LC3-II puncta" to "LC3 puncta" and specify which LC3 family member was monitored (LC3A, LC3B or LC3C).

Reviewer #2 (Remarks to the Author):

The Authors have addressed all comments from this reviewer

Reviewer #3 (Remarks to the Author):

Thank you for your revision. I still have one minor comment.

1) It would be crucial to show that the endogenous proteins also interact (AMPK-CyclinY-CDK16). Is this interaction autophagy regulated? Indeed, immunoprecipitation using co-IP buffer may be difficult; however, the authors can show immunofluorescence (with/without autophagy-stimulation).

Reviewer #4 (Remarks to the Author):

In their answer to my comment 10 the authors write that they quantified the blots in Fig 4A. However, there are no quantifications added to Fig. 4A in the revised version of the manuscript. These should be added. Otherwise, I am satisfied with the revisions made and answers provided to my comments/questions.

Rebuttal Dohmen et al., “AMPK-dependent activation of the Cyclin Y/CDK16 complex controls autophagy” (NCOMM-18-28338A)

First of all, we appreciate the comments by the four reviewers who have carefully read our revised manuscript and suggested a few additional modification / experiments. We have addressed all the comments point-by-point below and think that we have been able to provide appropriate answers and thus improved our manuscript.

Reviewers' comments:

Reviewer #1 (Remarks to the Author):

The authors provided sufficient explanations to address the majority of my concerns. However, a few open issues remain.

Add 2) Figure 4A-C and S3C indeed show that the CDK16's kinase activity is required but these settings are always overexpression approaches. Reconstituting CDK16 KO MEFs with CDK16-K194R would allow to strengthen the authors finding at near endogenous levels and more physiological conditions. Since the authors already used this reconstitution strategy for wild-type CDK16, it should be quite straightforward to expand this to the mutant.

We now used a lentiviral system with doxycycline inducible expression of GFP-CDK16 wt and a kinase dead mutant of CDK16 (KR). This system allowed us to adjust the expression of GFP-CDK16 wt and KR in the reconstituted CDK16^{-/-} MEFs to endogenous levels of Cdk16 in the CDK16^{+/+} MEFs. In these cells the GFP-CDK16 KR mutant was compared to the GFP-CDK16 wt and we found that the mutant CDK16 was unable to rescue the autophagic defect (new Figure S4F).

Add 5) In Figure S3A/B the authors monitored WIPI1. However, I asked to expand this assay to WIPI2 as this is one of the gold standards for autophagosome precursor structures.

To address this comment, we have performed endogenous staining of Wipi2b in the CDK16^{+/+} and CDK16^{-/-} MEFs with and without starvation (Figure S4A). The Wipi2b dot formation is quantified in Figure S4B. The reduction of Wipi2b dots in the CDK16^{-/-} MEFs in comparison to the CDK16^{+/+} MEFs is similar to the observations we made in the experiments in which we analyzed exogenous GFP-WIPI1 (new Figure S4C and D).

Besides, LC3-II refers to an immunoblot readout. Please change "LC3-II puncta" to "LC3 puncta" and specify which LC3 family member was monitored (LC3A, LC3B or LC3C).

Thank you very much for this commend. We have revised our text accordingly and changed LC3-II to LC3 when puncta are analyzed. Moreover, we specified the monitored LC3 form. We used the rabbit polyclonal anti-LC3B antibody from Cell Signaling Technology (#2775) for western blots (now indicated in the figure) and the mouse monoclonal LC3 antibody 4E12 from MBL (#M152-3) for immunofluorescence staining. This antibody recognizes LC3A and LC3B according to the data sheet. Now indicated in the figure legends.

Reviewer #2 (Remarks to the Author):

The Authors have addressed all comments from this reviewer

Thank you very much for your positive conclusion.

Reviewer #3 (Remarks to the Author):

Thank you for your revision. I still have one minor comment.

1) It would be crucial to show that the endogenous proteins also interact (AMPK-CyclinY-CDK16). Is this interaction autophagy regulated? Indeed, immunoprecipitation using co-IP buffer may be difficult; however, the authors can show immunofluorescence (with/without autophagy-stimulation).

We thank the reviewer for this comment. The interaction between endogenous Cyclin Y and AMPK by using Co-IP was not successful. However, using Co-IP experiments we were able to detect an interaction between the Cyclin Y S326A mutant and AMPK (new Figure S3B). These findings suggest that only a non-phosphorylatable mutant of Cyclin Y can bind stably enough to AMPK. An explanation for this might be that the enzyme-substrate interaction is too transient to be detected in cell lysates, whereas a non-phosphorylatable mutant most likely has a slower off-rate because of lack of modification.

However, we were able to detect the interaction of the endogenous proteins by antibody staining as suggested by the reviewer. In order to achieve high specificity, we used proximity ligation assays. We observe a specific signal for the interaction between Cyclin Y and AMPK β only in the presence of antibodies against both Cyclin Y and AMPK. Interaction increases slightly but significantly after EBSS treatment or stimulation of AMPK by AICAR/A769662.

Together these findings support the conclusion that AMPK is directly modifying Cyclin Y in cells.

Reviewer #4 (Remarks to the Author):

In their answer to my comment 10 the authors write that they quantified the blots in Fig 4A. However, there are no quantifications added to Fig. 4A in the revised version of the manuscript. These should be added. Otherwise, I am satisfied with the revisions made and answers provided to my comments/questions.

We apologize for not adding the quantifications in Figure 4a, which we have now added (Figure 4A).